# An extended trajectory-mechanics approach for calculating the path of a pressure transient: Traveltime tomography

Donald W. Vasco[1], Joseph Doetsch[2], and Ralf Brauchler[3]

[1]Energy Geosciences Division/Building 74, Lawrence Berkeley National Laboratory, 1 Cyclotron Road, Berkeley, CA 94720
[2]Department of Earth Sciences, ETH Zurich, Zurich, Switzerland
[3]Waste Disposal and Hydrogeology, AF-Consult Switzerland Ltd, Täfernstrasse 26, Baden, Switzerland CH-5405
**Correspondence:** D. W. Vasco (dwvasco@lbl.gov)

**Abstract.** The application of a technique from quantum dynamics to the governing equation for hydraulic head leads to a trajectory-based solution that is valid for a general porous medium. The semi-analytic expressions for propagation path and velocity of a change in hydraulic head form the basis of a traveltime tomographic imaging algorithm. An application of the imaging algorithm to synthetic arrival times reveals that a crosswell inversion based upon the extended trajectories correctly reproduces the magnitude of a reference model, improving upon an existing asymptotic approach. An inversion of hydraulic head arrival times from crosswell slug tests at the Widen field site in northern Switzerland captures a general decrease in permeability with depth, in agreement with previous studies, but also indicates the presence of a high permeability feature in the upper portion of the crosswell plane.

## 1 Introduction

Understanding the spatial variation in subsurface flow properties is important for many applications, such as groundwater extraction and storage, hydrocarbon production, geothermal energy generation, and waste water disposal. Advanced production processes like hydraulic fracturing require the development of high-resolution reservoir models necessary to capture the influence of the fractures [*Zhang et al.*, 2014; *Fujita et al.*, 2015]. Often there are very few observations with which to infer such properties, typically measurements from a few wells intersecting a formation of interest. However, developments such as crosswell transient pressure testing [*Hsieh et al.*, 1985; *Paillet*, 1993; *Karasaki et al.*, 2000;] and hydraulic tomography [*Tosaka et al.*, 1993; *Gottlieb and Dietrich*, 1995; *Butler et al.*, 1999; *Yeh and Liu*, 2000; *Vasco and Karasaki*, 2001; *Bohling et al.*, 2002, 2007; *Brauchler et al.*, 2003, 2010, 2011, 2013; *Zhu and Yeh*, 2006; *Illman et al.*, 2007, 2008; *Frienen et al.*, 2008; *Bohling*, 2009; *Cardiff et al.*, 2009, 2011, 2013a; 2013b; *Huang et al.*, 2011; *Sun et al.*, 2013; *Paradis et al.*, 2015, 2016], have improved the ability to resolve two- and three-dimensional variations in hydraulic properties. New techniques, including fiber optic temperature and pressure observations, and geophysical observations sensitive to pressure changes [*Yeh et al.*, 2008; *Rucci et al.*, 2010; *Marchesini et al.*, 2017], will further improve spatial and temporal coverage and generate large data sets. Finally, the joint interpretation and inversion of geophysical and hydrological data leads to better constrained imaging of flow properties [*Rubin et al.*, 1992; *Hyndman et al.*, 1994; *Hyndman et al.*, 2000; *Vasco et al.*, 2001; *Vasco*, 2004; *Kowalsky et al.*,

2004; *Day-Lewis et al.*, 2006; *Brauchler et al.*, 2012; *Lochbühler et al.*, 2013; *Soueid Ahmed et al.*, 2014: *Ruggeri et al.*, 2014: *Jimenez et al.*, 2015; *Binley et al.*, 2015; *Linde and Doetsch*, 2016 ].

The characterization of complicated aquifer and reservoir models using sizable data sets points to the need for robust and efficient approaches for modeling pressure propagation. To this end, there are a number of approaches that aim to reduce the computational burden and data handling requirements associated with hydraulic tomography. For example, there are methods that reduce the governing equation to a simpler form for the moments of the transient head or pressure variation [*Li et al.*, 2005; *Yin and Illman*, 2009; *Zhu and Yeh*, 2006]. There are also approaches for the analysis of sinusoidal and oscillatory pumping tests that are based upon the phase shifts and amplitude differences between observed and calculated pressure variations, using these phase shifts to infer properties between two wells [*Bernabe et al.*, 2005; *Black and Kipp*, 1981; *Cardiff et al.*, 2013b; *Kuo*, 1972; *Rasmussen et al.*, 2003; *Renner and Messar*, 2006]. Another technique relies upon a measure of the arrival time of a pressure pulse or disturbance as a basis for transient traveltime imaging or tomography [*Vasco et al.*, 2000; *Kulkarni et al.*, 2001; *Brauchler et al.*, 2003, 2007, 2010, 2011, 2013; *He et al.*, 2006; *Hu et al.*, 2011; *Vasco and Datta-Gupta*, 2016]. Finally, there are methods that attempt to find lower-dimensional representations of the model or of the matrices describing the forward and inverse problems. These methods include principal component analysis [*Lee and Kitanidis*, 2014;], Karhunen-Loeve expansions [*Zha et al.*, 2018], and reduced-order models [*Liu et al.*, 2013].

There are at least three advantages associated with the use of travel times, an alternative to the direct treatment of the entire transient head or pressure waveforms. First, the arrival of the early onset of the transient pressure pulse can be much sooner than the time at which steady-state conditions are achieved. Thus, crosswell slug tests can be conducted rapidly, facilitating improved spatial coverage. Second, the relationship between such travel times and hydraulic diffusivity is quasi-linear and convergence to a solution is not as sensitive to the initial model as it is for the direct inversion of transient pressure waveforms [*Cheng et al.*, 2005]. Third, the interpretation and reduction of transient head or pressure waveform data can be more complicated due to the sensitivity of amplitudes to various factors such as the packer coupling, the calibration of the receiver transducers, and the conditions surrounding the borehole.

Previous trajectory-based formulations of pressure arrival time tomography relied upon an asymptotic approach that assumes smoothly-varying properties [*Vasco et al.*, 2000; *Brauchler et al.*, 2003, 2007; *He et al.*, 2006; *Vasco*, 2008; *Vasco and Datta-Gupta*, 2016]. This assumption is certainly violated in many commonly encountered situations, such as a layered sedimentary environment and in the presence of faults or fractures. Here we apply a newly developed trajectory-based technique for travel time tomography that dispenses with the assumption of smoothly-varying properties, enlarging its range of validity to any model that may be treated using a numerical simulator [*Vasco*, 2018; *Vasco and Nihei*, 2019]. The semi-analytic approach provides insight into factors controlling the propagation of a pressure transient in a complex porous medium. As shown here, the expression for the trajectories may form the basis for efficient sensitivity computations. These sensitivities are particularly useful in inverting transient pressure propagation times and in hydraulic travel time tomography. All of the the sensitivities required for the interpretation of a pressure test can be obtained in a single numerical simulation of the test. We apply the method to crosswell hydraulic tomographic imaging, considering both synthetic and field pressure arrival times.

## 2 Methodology

In this section we describe our iterative algorithm for updating an aquifer model in order to improve the fit to a set of observed arrival times. We shall only discussion the elements of the derivation of *Vasco* [2018], as well as a perturbation technique, that are essential for understanding the inversion procedure. The approach involves a number of steps, beginning with the equation governing the transient variation in hydraulic head, and ending with a linear system of equations to be solved for the aquifer parameters. As an overview, the major steps of the methodology are shown schematically in Figure 1. The approach is an off-shoot of trajectory-based techniques developed in quantum dynamics for the study of large chemical systems [*Wyatt*, 2005; *Liu and Makri*, 2005; *Goldfarb et al.*, 2006; *Garashchuk*, 2010; *Garashchuk and Vazhappilly*, 2010; *Garashchuk et al.*, 2011; *Gu and Garashchuk*, 2016]. As shown in *Vasco* [2018], the trajectory mechanics treatment leads to a set of coupled ordinary differential equations that may be solved numerically, as is done in quantum mechanics. However, one can take advantage of existing numerical simulators to compute one of the unknown vector fields, reducing the system to a single set of equations for the trajectory [*Vasco*, 2018]. The result of this analysis is a semi-analytic expression for the path of a transient pulse. This expression, along with a perturbation technique, provides a basis for an efficient method for imaging spatial variations in hydraulic diffusivity in the subsurface, a form of travel time tomography. We illustrate the procedure with applications to both synthetic and observed arrival times in the section that follows this description.

### 2.1 Governing equation and trajectory calculations

We begin with the equation governing the evolution of a transient variation in hydraulic head $h(\mathbf{x},t)$ [L] as a function of space $\mathbf{x}$ and time $t$, adopting the form of the governing equation presented in *de Marsily* [1986, p. 109]

$$\nabla \cdot (\mathbf{K} \cdot \nabla h) = \zeta \frac{\partial h}{\partial t} \tag{1}$$

where $\mathbf{K}$ is the hydraulic conductivity [L/T], a symmetric tensor, and $\zeta$ is the specific storage coefficient with dimensions of length$^{-1}$ [1/L]. The specific storage coefficient depends upon the total porosity of the medium, the isothermal compressibility of the liquid, the compressibility of the solid constituents, and the compressibility of the porous matrix, as discussed in *de Marsily* [1986, p. 109].

From this point on we shall assume that the hydraulic head has been normalized by dividing both sides of equation (1) by a constant reference head value $h_0$ [L]. We will still use the variable $h(\mathbf{x},t)$ for the normalized head, which is now unitless. An expression for the trajectory associated with the propagation of a transient fluid front follows from substituting the exponential representation

$$h(\mathbf{x},t) = e^{-S(\mathbf{x},t)} \tag{2}$$

into the governing equation (1) for hydraulic head. Because we can choose the reference location such that the hydraulic head is always positive, equation (2) is well defined and can always be solved for $S$. Upon substituting for $h(\mathbf{x},t)$ in equation (1),

the resulting equation for $S(\mathbf{x},t)$, known as the phase, may be written as

$$\frac{\partial S}{\partial t} + \mathbf{v} \cdot \nabla S = \frac{1}{\zeta} \nabla \cdot (\mathbf{K} \cdot \mathbf{p}).$$  (3)

The vector $\mathbf{p}$ [1/L] is the spatial gradient of the phase

$$\mathbf{p} = \nabla S,$$  (4)

and $\mathbf{v}$ [L/T] is a velocity vector given by

$$\mathbf{v} = \mathbf{p} \cdot \frac{\mathbf{K}}{\zeta}.$$  (5)

Note that equation (3) has the form of a traveling front with a velocity that depends upon the vector $\mathbf{p}$ and the medium properties $\zeta$ and $\mathbf{K}$. As shown in *Vasco* [2018], the partial differential equation (3) is equivalent to the system of ordinary differential equations

$$\frac{d\mathbf{x}}{dt} = \frac{1}{\zeta} \mathbf{p} \cdot \mathbf{K}$$  (6)

$$\frac{d\mathbf{p}}{dt} = \nabla \left[ \frac{1}{\zeta} \nabla \cdot (\mathbf{K} \cdot \mathbf{p}) \right].$$  (7)

One can solve the two ordinary differential equations for the trajectory $\mathbf{x}$ and the vector $\mathbf{p}$ [*Cash and Carp*, 1990; *Press et al.*, 1992; *Wyatt*, 2005]. An alternative approach is to use a reservoir simulator to calculate $h(\mathbf{x},t)$ and then use equation (2) and (4) to determine $\mathbf{p}$ from the hydraulic head

$$\mathbf{p} = -\nabla \ln h = -\frac{\nabla h}{h}.$$  (8)

Substituting for $\mathbf{p}$ in equation (6) gives an expression for the trajectory in terms of $h(\mathbf{x},t)$

$$\frac{d\mathbf{x}}{dt} = \mathbf{v} = -\frac{\mathbf{K}}{\zeta} \cdot \frac{\nabla h}{h}.$$  (9)

We use the numerical simulator TOUGH2 [*Pruess et al.*, 1999] to calculate the pressure and head changes and then use equation (9) to find the trajectories.

## 2.2  Semi-analytic sensitivities

A primary application of the trajectories described above will be to estimate flow properties between boreholes via hydraulic tomographic imaging. In this procedure a series of pumping tests are conducted in isolated segments of one borehole. During each test a rapid injection is used to generate a transient fluid pressure pulse that propagates to pressure sensors in an adjacent well. For an impulsive source, the time at which the peak pressure is observed in the adjacent borehole is defined as the arrival time. For the inverse problem we determine the flow properties from the arrival times observed in isolated sections of the monitoring well. In order to solve the inverse problem we must relate the travel time of the pressure pulse to the hydraulic properties of the medium.

Our approach to the solution of the non-linear inverse problem will be iterative in nature. That is, in order to estimate flow properties we begin with an initial model and progressively update it, solving the forward problem of reservoir simulation at each step. We shall need model parameter sensitivities, the partial derivatives of each observation with respect to changes in each of the model parameters [*Jacquard and Jain*, 1965], for every iterative update. We will be interested in transient pressure arrival times that are defined as the time at which the peak of a pressure pulse is observed at a measurement point. Expression (9) forms the basis for our sensitivity estimates. The only non-zero component of the velocity vector $\mathbf{v}$ is along the trajectory $\mathbf{x}(t)$, and it is given by the magnitude of the vector, denoted by $v$. Integrating equation (9) along the path $\mathbf{x}(t)$ we have

$$T = \int_{\mathbf{x}} \frac{dx}{v}, \tag{10}$$

where $x = |\mathbf{x}|$ is the distance along the path $\mathbf{x}$. One could relate perturbations in the arrival time of a pressure pulse with respect to changes in the velocity but this will lead to a sensitivity that varies as $v^{-2}$. This will magnify the influence of any variations in velocity along the trajectory, potentially leading to instabilities in the inversion. Formulating the inverse problem in terms of the slowness, s [T/L], given by

$$s = \frac{1}{v}, \tag{11}$$

eliminates this problem and leads to

$$T = \int_{\mathbf{x}} s(x)dx, \tag{12}$$

an integral relationship where the non-linearity is contained entirely within the definition of the path of integration. That is, according to equation (9), the path of integration $\mathbf{x}$ depends upon $\mathbf{v}$ and hence $s(x)$.

Model parameter sensitivities, in this case relating small changes in the slowness along the trajectory, $\delta s(\mathbf{x})$, to changes in the travel time of a transient pressure pulse, follow from a perturbation argument. Specifically, we consider a perturbation of the slowness with respect to a background model $s_o(\mathbf{x})$

$$s(\mathbf{x}) = s_o(\mathbf{x}) + \delta s(\mathbf{x}) \tag{13}$$

where $\delta s$ is assumed to be small. There is a corresponding small change, $\delta T(\mathbf{x})$, in the travel time from a source location to an observation point

$$T(\mathbf{x}) = T_o(\mathbf{x}) + \delta T(\mathbf{x}). \tag{14}$$

Substituting the perturbed forms of $s(\mathbf{x})$ and $T(\mathbf{x})$, given by equations (13) and (14), into the expression (12) produces

$$T_o(\mathbf{x}) + \delta T(\mathbf{x}) = \int_{\mathbf{x}} s_o(\mathbf{x})dx + \int_{\mathbf{x}} \delta s(\mathbf{x})dx, \tag{15}$$

where the integration is along a perturbed path $\mathbf{x} = \mathbf{x}_o + \delta \mathbf{x}$. It has been shown that perturbations in the path lead to terms that are second order in $\delta s$. Thus, in computing the sensitivities, which are first order in $\delta s$, we can neglect perturbations in the

trajectory due to perturbations in $s$. Therefore, we can integrate along the path calculated for the current or background model, denoted by $\mathbf{x}_o$. Because the traveltime in the background model, $T_o$, is the integral of the background slowness function $s_o(\mathbf{x})$ along the trajectory, the initial terms on each side of equation (15) cancel and we are left with

$$\delta T = \int\limits_{\mathbf{x}_o} \delta s(x)dx, \tag{16}$$

relating perturbations in the slowness, $\delta s$, along the trajectory to perturbations in the arrival time, $\delta T$.

In order to update the model and the head or pressure field using a numerical simulator, we shall need to map the updated slowness estimates into the reservoir model parameters $\zeta$ and $\mathbf{K}$. This cannot be done in a unique fashion and requires additional information or assumptions. Here, we will assume that the permeability tensor is isotropic, so that it is of the form $\mathbf{K} = K\mathbf{I}$, where $K$ is the scalar permeability and $\mathbf{I}$ is the identity matrix with 1's on the diagonal and 0's elsewhere. If the inversion is part

of a joint inversion of several data types it might be possible to solve for $\zeta$ or $K$ using other information, such as geophysical observations. In some formations, such as a clean sand, it might be possible to relate the permeability to the porosity, and to solve for the porosity uniquely in terms of the slowness. Alternatively, since the porosity typically has a much smaller range of variation than permeability does, one might assume that the permeability dominates variations in $s$, and hence solve for an effective permeability, lumping both changes in $\zeta$ and permeability into changes in $K$. It is evident from equation (9) that one

has to correct the estimates for variations in hydraulic head. As shown below, we use the output of the numerical simulator, based upon the current reservoir model, for this correction.

## 2.3  Comparison with existing asymptotic methods

Several trajectory-based methods for pressure arrival time tomography [*Vasco et al.*, 2000; *Brauchler et al.*, 2003; *He et al.*, 2006; *Hu et al.*, 2011; *Vasco and Datta-Gupta*, p. 131, 2016] utilize a high-frequency asymptotic solution to the diffusion

equation. A major assumption of such solutions is that the pressure variation is rapid in time [*Virieux et al.*, 1994] or that the dominant frequencies in a Fourier transform of the trace are high. Equivalent results can be obtained if we assume that the medium properties are smoothly-varying in comparison with the length scale associated with the propagating pressure transient or that parameters take on values in a particular range [*Cohen and Lewis*, 1967]. In that case we can neglect the divergence term on the right-hand-side of equation (3) and it reduces to an eikonal equation

$$\frac{\partial S}{\partial t} + \frac{K}{\zeta}\nabla S \cdot \nabla S = 0, \tag{17}$$

where we have made use of equations (4) and (5). There are efficient fast-marching methods for solving the eikonal equation [*Podvin and Lecomte*, 1991; *Sethian*, 1999; *Osher and Fedkiw*, 2003], that are applicable to modeling transient pressure propagation in high resolution reservoir models [*Zhang et al. 2014*, *Fujita et al.*, 2015]. The eikonal equation is equivalent to a system of ordinary differential equations, the ray equations, defining the path of the transient pulse and the spatial variation of

the phase [*Courant and Hilbert*, 1962].

From the high frequency asymptotic solution and the ray equations *Vasco et al.* [2000] derived a semi-analytic expression, in which the square root of the peak arrival time is given by the line integral along the trajectory $\mathbf{x}_{eikonal}$ defined by the eikonal

equation,

$$\sqrt{T_{peak}} = \int_{\mathbf{x}_{eikonal}} \varphi dr \tag{18}$$

where

$$\varphi = \frac{1}{6}\sqrt{\frac{\zeta}{K}} \tag{19}$$

has units of $\sqrt{T}/L$, $\mathbf{x}_{eikonal}$ signifies the trajectory resulting from the solution of the eikonal equation (17), and $r$ is the distance along the trajectory. Equation (18) is a nonlinear relationship between the travel time $T_{peak}$ and $\varphi$ because the path $\mathbf{x}_{eikonal}$ depends upon the spatial variation of $\zeta$ and $K$. As in the previous sub-section, we can linearize the relationship by assuming a background model and considering perturbations, or small changes, with respect to the background model. Because the perturbations in the path $\mathbf{x}_{eikonal}$ are second order in perturbation of $\varphi$, we can write the perturbed expression as

$$\delta\sqrt{T_{peak}} = \int_{\mathbf{x}_0} \delta\varphi dr \tag{20}$$

where $\delta\sqrt{T_{peak}}$ is the perturbation in the square root of the travel time and $\mathbf{x}_0$ is the trajectory in the background medium.

In *Vasco* [2018] the limitations of the high frequency asymptotic approach are discussed and illustrated. In particular, it is shown that for abrupt boundaries and sharp layers, the trajectories calculated using the eikonal equation bend too strongly into high permeability regions of a half-space or layer. This leads to deviations in the trajectories from regions with high model parameter sensitivity, and the potential for errors when updating a simulation model. In the next section we will explore these limitations in the context of hydraulic tomography, using both synthetic and experimental data.

## 2.4 A Linearized and Iterative Approach for Tomographic Imaging

A reservoir model is typically defined over a two- or three-dimensional grid that is used by a numerical reservoir simulator. For such a discrete model with properties defined on a grid of cells, and where one assumes constant values within each cell of the model, we can break up the path integrals (16) and (20) into sums over all of the grid blocks intersected by the trajectories. For the integral (16) the discrete sum is given by

$$\delta T = \sum_{i=1}^{N} l_i \delta s_i, \tag{21}$$

where $\delta s_i$ is the perturbation of $s$ in the $i$-th grid block, and $l_i$ is the length of the trajectory $\mathbf{x}_o$ in that grid block. Equation (21) constitutes a linear constraint on the perturbations of $s$ in the sampled grid blocks of the model, those intersected by the path $\mathbf{x}_0$. By considering a number of sources and receiver pairs, for example from a sequence of cross well slug tests, we arrive at a system of linear equations relating the perturbations in $s$ to perturbations in the observed arrival times. We may write the system as a matrix equation

$$\delta\mathbf{T} = \mathbf{M}\delta\mathbf{s}, \tag{22}$$

where $\delta\mathbf{T}$ is a vector of travel time residuals, $\mathbf{M}$ is a matrix containing the path lengths in each grid block intersected by one or more trajectories, and $\delta\mathbf{s}$ is a vector whose elements consist of the perturbations in each grid block, $\delta s_i$ or $\delta\varphi_i$ for the $i$-th block.

In the iterative, linearized inversion scheme that we shall adopt here, we start with an initial model, perhaps derived from well logs, denoted by $\zeta_0$ and $K_0$, and calculate the background values of $v$ or $\varphi$. Depending upon the method, we either conduct a reservoir simulation and use equation (9) to derive the trajectories, or solve the eikonal equation (17) and calculate $\mathbf{x}_{eikonal}$ as in *Vasco et al.* [2000]. This allows us to compute the trajectories and the lengths in each grid block and to construct the elements of the matrix $\mathbf{M}$. In order to update the reservoir model we need to relate the updated field $s$ to the model parameters $\zeta$ and $K$. Recall that we can only resolve the ratio $\zeta/K$ and we cannot distinguish increases in $\zeta$ from decreases in $K$ and vice-versa. In this example only the permeability varies, so we fix $\zeta$ and only solve for changes in permeability. If $\zeta$ also varies then we can only find an effective permeability variation that will contain the effects of any variation in $\zeta$. The relationship follows from equation (9) and is given by

$$K = \frac{\zeta}{s|\nabla \ln h|}. \tag{23}$$

Because the head field $h(\mathbf{x}, t)$ is present in the integral expression, we need to recalculate this field at each iteration. But that calculation is already required in order to update the trajectory $\mathbf{x}(t)$.

Due to errors in the data and modeling approximations, we do not expect that the system of equations (22) will have an exact solution. Thus, we seek a least squares solution in which the sum of the squares of the residuals is minimized. Furthermore, due to resolution and uniqueness issues, a direct least squares solution of (22) will probably be unstable and small errors will lead to large changes in the estimates of $\delta\mathbf{s}$ [*Menke*, 2012; *Parker*, 1994]. Therefore we introduce regularization or penalty terms in order to stabilize the inverse problem. The penalty terms seek to minimize the norm of the model update, and to minimize the roughness of the updates, as measured by the difference operators that mimic the second spatial derivatives of the model, the model Laplacian [*Menke*, 2012]. The function that we are minimizing, $\Pi(\delta\mathbf{s})$, is the sum of the squares of the residuals, the weighted model norm and the weighted model roughness:

$$\Pi(\delta\mathbf{s}) = (\delta\mathbf{T} - \mathbf{M}\delta\mathbf{s})^t \cdot (\delta\mathbf{T} - \mathbf{M}\delta\mathbf{s}) + w_n \delta\mathbf{s}^t \cdot \delta\mathbf{s} + w_r (\mathbf{L}\delta\mathbf{s})^t \cdot (\mathbf{L}\delta\mathbf{s}), \tag{24}$$

where $\mathbf{L}$ is a matrix operator that mimics the second spatial derivative of the model, $w_n$ is the model norm weight, and $w_r$ is the model roughness weight. Note that in equation (24) we are weighting all the data uniformly. It is possible to include a covariance matrix in order to account for correlations between observations and variations in data quality [*Tarantola*; 2005]. Minimizing the quadratic function (24) with respect to the model parameters leads to a linear system of equations for $\delta\mathbf{s}$

$$\left[\mathbf{M}^t\mathbf{M} + w_n\mathbf{I} + w_r\mathbf{L}^t\mathbf{L}\right]\delta\mathbf{s} = \mathbf{M}^t\delta\mathbf{T}. \tag{25}$$

The penalized least squares problem is solved for the perturbations, $\delta\mathbf{s}$, using the Least Squares QR algorithm (LSQR) proposed by *Paige and Saunders* [1982]. With the solution in hand we then update the reservoir model. Because the high frequency asymptotic method only requires $\varphi$, we do not need to convert back to the flow parameters $\zeta$ and $K$. Therefore, we can update

the model, solve the updated eikonal equation, recompute the residuals, retrace the trajectories, calculate the sensitivities, and continue the process until the misfit to the travel times is reduced sufficiently. For the extended trajectory method we can use equation (23) to transform from $s$ to $K$ before updating the reservoir model and conducting another numerical simulation.

The linearized expression (25) also provides a basis for the assessment of a solution to the inverse problem, that is, the calculation of model parameter resolution and uncertainty [*Parker*, 1994; *Aster et al.*, 2005; *Menke*, 2012]. Model parameter resolution estimates can be particularly useful in understanding spatial averaging and non-uniqueness in hydrological inverse problems [*Vasco et al.*, 1997; *Bohling*, 2009; *Paradis et al.*, 2016]. We can define model parameter resolution very simply in terms of the generalized inverse $\mathbf{G}^{\dagger}$, obtained from equation (25) by formally inverting the matrix on the left-hand-side

$$\mathbf{G}^{\dagger} = \left[\mathbf{M}^t\mathbf{M} + w_n\mathbf{I} + w_r\mathbf{L}^t\mathbf{L}\right]^{-1}\mathbf{M}^t. \tag{26}$$

Hence, the parameters estimates for a given iteration, denoted by $\delta\hat{\mathbf{s}}$, are a linear function of the observations

$$\delta\hat{\mathbf{s}} = \mathbf{G}^{\dagger}\delta\mathbf{T}. \tag{27}$$

Using equation (22) to replace $\delta\mathbf{T}$ by $\mathbf{M}\delta\mathbf{s}$ gives a relationship between the estimated parameters and the 'true' parameters,

$$\delta\hat{\mathbf{s}} = \mathbf{G}^{\dagger}\mathbf{M}\delta\mathbf{s} = \mathbf{R}\delta\mathbf{s}, \tag{28}$$

where $\mathbf{R}$ is the resolution matrix [*Menke*, 2012], with rows that are coefficients describing the averaging that occurs in estimating a parameter. We can also make use of the linear relationship between the residuals and the model parameter updates, given by equations (22) and (27), to estimate an a posteriori model parameter covariance matrix $\mathbf{C}_{ss}$ in terms of the covariance matrix of the errors associated with the observed traveltimes. In particular, if the data errors are Gaussian, characterized by the data covariance matrix $\mathbf{C}_{TT}$ then the model parameter covariance matrix, may be written in terms of the generalized inverse and the data covariance matrix

$$\mathbf{C}_{ss} = \mathbf{G}^{\dagger}\mathbf{C}_{TT}\left(\mathbf{G}^{\dagger}\right)^{t}, \tag{29}$$

a consequence of the linear nature of the problem and the properties of the Gaussian distribution [*Menke*, 2012].

## 3 Applications

Crosshole hydraulic traveltime tomography and crosswell slug tests are valuable approaches for imaging spatial variations in flow properties [*Paillet*, 1993; *Yeh and Liu*, 2000; *Vasco and Karasaki*, 2001; *Bohling et al.*, 2002; *Butler et al.*, 2003; *Brauchler et al.*, 2007; *Brauchler et al.*, 2010; *Brauchler et al.*, 2011]. Such tests can resolve features between boreholes, similar to crosswell geophysical imaging, and are directly sensitive to flow properties. In this section we set up a synthetic hydraulic tomographic test, roughly based upon a field experiment at the Widen site in Switzerland. Following that, we analyze data from the actual field experiment, using them to image the spatial variations of permeability between two shallow boreholes.

## 3.1 Synthetic hydraulic tomography test case

The overall setup of the test example is shown in Figure 2, along with the reference model. A set of sources in each well, denoted by filled squares and open circles, transmit transient pressure signals to various receivers located in the adjacent borehole. The reference distribution, a three-dimensional permeability model with a dominantly vertical variation in properties, was generated

stochastically. That is, a uniform number generator was used to derive permeability multipliers between 1 and 12 for each layer in the model. A uniform random variation of 50% was introduced within each layer and this variation was smoothed using a three point moving window. The model extends an additional 5 meters in the $x$, $y$, and $z$ directions, beyond the boundaries of the plane defined by the crosswell survey.

The reservoir simulator TOUGH2 [*Pruess et al.*, 1999] was used to model the complete set of crosswell slug tests that

comprised the full synthetic experiment. The computations were conducted using a three-dimensional mesh with constant pressure boundary conditions, simulating a 300 s transient pressure test for each source. This interval provided enough time for any head variation to propagate from a source to the receivers due to the high background permeability of $5.0 \times 10^{-10}$ m$^2$. The large background permeability allowed us to match the rapid pulse propagation between the boreholes that was observed during the actual Widen field experiment described below. The initial conditions where a constant pressure of 0.616 MPa and

a uniform temperature of $20^o$ C. The source-time function was defined by a jump in flow rate followed by an exponentially decreasing rate. The transient arrivals were defined as the time at which the rate of change in the pressure or head reached a maximum value. A set of synthetic arrival times were calculated using TOUGH2 simulations and then used as a test data set for the imaging algorithm described above. Uniform random deviates, with maximum variations of 5% of the arrival time, were generated using a pseudo-random number generator and added to the TOUGH2 calculated travel times.

In order to image the permeability variations between the boreholes we conducted a series of linearized inversion steps, where we solve the system of equations (25) at each step. The starting model is a uniform half-space with a permeability of $5.0 \times 10^{-10}$ m$^2$. The model extends from 0.0 to 15.0 m laterally and from 0.0 to 15.0 m in the vertical direction. We represent the crosswell area using a 33 by 33 grid of cells with a block size of 0.45 m and embed this into a 15 m (in the $z$ direction) thick three-dimensional model. Each of the injection events was simulated for 300 s, even though the pressure transient arrived

at the observation points just a few seconds after the beginning of the test. The pressure field from the simulation was used to compute the trajectories, using the expression (9) for the tangent vector, and integrating it to construct the entire path. The arrival times were calculated using both the eikonal equation (17) and by post-processing the simulation results to estimate the arrival time of the propagating transient as it reached each observation point. The linearized iterative algorithm, where equation (25) is solved at each step, was applied using both the eikonal equation and the extended trajectory approach to compute the

sensitivities in **M**.

The regularization weightings for each approach, $w_n$ and $w_r$ in equation (25), were estimated by trial and error. In particular, a series of inversions were conducted for various values of $w_n$ and $w_r$ and a balance was struck between satisfying the data and minimizing the model norm and roughness. For the eikonal-based inversion the misfit was calculated using travel times from the eikonal equation. For the new approach based upon the extended trajectories the travel time misfit was calculated using the

pressures from the numerical simulator TOUGH2. The norm and roughness weights for the iterative eikonal inversion were $w_r$ =0.15 and $w_n$=0.15. For the inversion utilizing the extended trajectories we set $w_r$ and $w_n$ equal to 0.1 and 0.5, respectively.

At each iteration we solve for a permeability multiplier, a factor that is multiplied by the background permeability of the uniform starting model to get the estimated permeability. A total of 10 iterations for the eikonal-based algorithm took 6 s while ten iterations for the extended trajectory approach took 129 minutes, illustrating the computational advantage provided by an inversion approach based upon the eikonal equation. In Figure 3 we plot the misfit reduction as a function of the number of iterations for both the high frequency inversion algorithm (Eikonal) and an inversion based upon the extended trajectories computed using equation (9). There is a large initial error reduction for both the inversion based upon the eikonal paths and the inversion utilizing the extended trajectories. However, as we continue updating the model and the size of the anomalies increase and the model becomes rougher, the error reduction for the two approaches diverge, and the eikonal-based updates no longer improve the fit when the reservoir simulator is used to calculate the arrival times. Note that the iterations do reduce the error calculated using the eikonal equation and the updated model, pointing to the differences between traveltime predictions made using a high frequency asymptotic approach and using the pressure equations. This highlights the fact that the eikonal equation becomes less accurate as the model starts to violate the assumptions of a smoothly-varying medium, an aspect supported by the results of *Vasco* [2018]. The misfit reduction associated with an iterative inversion algorithm utilizing the extended trajectories is also shown in Figure 3. In this case the reduction is essentially monotonic and the final error is much less than that of the eikonal-based approach. The number of iterations required to attain convergence depends upon a several factors. Two important elements are how close the initial model is to the final solution in model space and the level of errors in the observations that are being fit, including modeling errors. For the synthetic case considered here the level of random noise in the simulated arrival times is only 5 %. However the modeling error becomes an issue when the asymptotic approach is no longer valid or because we assume that the permeability outside of the crosswell plane is uniform.

The final updated high frequency solution, plotted in Figure 4, contains higher permeabilities between about 5.5 and 7.0 m. However, the amplitude of the permeability multiplier is less than that of the reference model (Figure 2). Furthermore, the amplitude of the high permeability feature at around 9.0 m is underestimated, perhaps due to its narrow width of less than a meter. The iterative inversion based upon the extended trajectories does image the two higher permeability zones seen in the reference model (Figure 2). The estimated amplitudes of the features appears to be closer to those of the reference model but it does overestimate the permeability of the lower feature and underestimates the permeability of the upper zone.

A better idea of the differences in the magnitude of the two solutions is conveyed in Figure 5, where we plot the depth variation of the reference, eikonal-based, and extended trajectory-based models. That is, we display the depth variation of the average of the two models, along with the upper and lower permeability multiplier values obtained in each depth interval. It is evident that the solution provided by a conventional imaging algorithm that uses the eikonal equation displays permeability changes with depth that are much smoother than the reference model. The extended approach does contain $K$ multipliers that are similar in size to the those of the reference model. Note that the exact locations of the high permeability features do vary in depth, deviating somewhat from the reference model shown in the left-most panel. This may be due to the wide, roughly 0.5 m, spacing of the source and receivers, and the low spatial resolution of pressure data in general [*Vasco et al.*, 1997].

Figure 6 provides more information regarding the misfit reductions for the inversions based upon the eikonal equation paths and the extended paths. It displays the calculated travel times plotted against travel times calculated using the reference model shown in Figure 2. Both the initial travel times, calculated using the homogeneous background model used to start the inversions, and the final travel times based upon the models obtained at the conclusion of the algorithms, are shown in the plots.

The initial travel time estimates are all larger than the actual values calculated using the reference model. This is to be expected because the largest anomalies are the approximately order-of-magnitude increases associated with the upper and lower high permeability layers in the model. The high permeability channels promote rapid pressure propagation between the boreholes. The eikonal equation-based algorithm does reduce the average of the calculated travel times but does not lead to good fits. The inversion based upon the extended trajectories produces relatively good fits to the reference travel times.

An important aspect of the inverse problem is an assessment of the resulting model parameters and estimates of their reliability. As noted in the Methodology section, the calculation of two key components of the model assessment: model parameter resolution and model parameter covariance or uncertainty, follow from the generalized inverse $\mathbf{G}^{\dagger}$ given by equation (26). As noted in the discussion surrounding equation (21), the coefficients required for construction of the sensitivity matrix are provided by trajectory-based, semi-analytic quantities, in particular by the ray lengths of the trajectories through each grid

block of the model. The formulas (28) and (29) provide the model parameter resolution and model parameter covariance, respectively. In Figure 7 we plot the diagonal elements of the resolution matrix and the standard errors associated with the estimates in Figure 4. The diagonal elements of the resolution matrix are plotted in the grid blocks to which they correspond. Because the rows of the resolution matrix are averaging coefficients that are normalized to have unit magnitude, if the diagonal element approaches 1 then the other elements approach 0. Therefore, diagonal values near 1 signify well resolved parameters

for those cells and little lateral averaging between nearby grid blocks. In general, the resolution for the test inversion is quite good and most parameters are well determined. Near the upper right corner the resolution does approach 0 due to the lack of sampling in that region. The resolution is highest in the central region, away from the edges of the model, due to the crosssing of trajectories in those areas. The estimated model parameter standard errors, also plotted in Figure 7, have a very different distribution, with larger values at the edges of the crosswell region where there are fewer crossing rays. We have scaled

the uncertainties by the magnitude of the estimated model parameters in order to plot them as percentages of the parameter estimates. The data errors were of the order of 10% of the magnitude of the travel time residuals that constitute the elements of the vector $\delta\mathbf{T}$ in equation (27). The lowest errors are in areas of high resolution, with the exception of the upper right-hand-corner where there is little ray coverage. In general, the errors are less than 5% of the magnitude of the estimated model parameters.

We end our treatment of the synthetic test with a discussion of some validation calculations, in which additional sources were introduced to mimic independent pumping tests. Two tests were simulated, with one source at the left edge of the model shown in Figure 2, at a height $Y$ of 9.9 m, and the other with a source in the right borehole at a height of 10.3 m. The travel times of transient pulses that propagate through the model, computed using the TOUGH2 simulator, are shown in Figure 2. Following that, TOUGH2 was used to calculate propagation times through the models shown in Figure 4. The respective arrival

35 times through the two models are plotted against the travel times for the reference model (Figure 8). The general trends of the

residuals do agree, with increasing calculated travel times following larger observed arrival times. For the eikonal-based model there are notable deviations from the $45^o$ line indicating perfect fit. The calculated travel times are systematically larger than the reference times. The estimates based upon the extended trajectory-based algorithm are much closer to the reference times than are the times from the eikonal approach.

## 3.2 The Widen field experiment

The Widen field site, adjacent to the Thur River in northern Switzerland (Figure 9), has been the subject of numerous geophysical and hydrological studies [*Lochb̈uhler et al.*, 2013]. The primary goal of the work at the Widen site is to understand the hydrologic, ecologic, and biochemical effects of river restoration. The geophysical and hydrological experiments focused upon a sandy gravel aquifer that is in contact with an unrestored section of the river [*Doetsch et al.*, 2010]. The area was penetrated by a number of boreholes and is relatively well characterized. Borehole cores revealed that the roughly seven-meter-thick sandy gravel aquifer is overlain by a silty sand layer and that it sits atop a thick impermeable clay aquitard. Early work at the site included individual and joint inversions of crosswell seismic, radar, and electrical resistance tomography for a zoned model [*Doetsch et al.*, 2010]. The model was consistent with the three-layer structure defined by the existing boreholes. This study was followed by several others, including a cross-hole ground-penetrating radar investigation [*Klotzsche et al.*, 2010], and three-dimensional electrical resistance tomographic (ERT) imaging of river infiltration into the site [*Coscia et al.*, 2011; 2012]. The three-dimensional ERT imaging indicates that the highest flow velocities occur in the middle of the aquifer while the lowest speeds are at the base of the sequence in clay and silt-rich gravels. A joint inversion of geophysical and hydrological data [*Lochbühler et al.*, 2013] between several well pairs was used to constrain spatial variations in reservoir storage and hydraulic conductivity. That study imaged the large-scale decrease in hydraulic conductivity with depth.

Crosswell slug interference tests, as described in *Brauchler et al.* [2010; 2011], were conducted at the site and are discussed in *Lochbühler et al.* [2013]. In such tests, a near instantaneous change in hydraulic head in a packed-off section of one well generates a fluid pressure transient in the surrounding region. Pressure transducers in isolated sections of a nearby well are used to measure the pulse that propagates between the wells. Both the travel time of the pulse and its amplitude can be used to infer hydraulic properties between the wells [*Vasco et al.*, 2000; *Brauchler et al.*, 2007; *Vasco*, 2008; *Brauchler et al.*, 2011]. Crosswell interference slug test were conducted at two well pairs at the Widen site, as described by *Lochbühler et al.* [2013]. The wells P2, P3, and P4 are roughly in a line that parallels the Thur river at a distance of 15 meters from the river bank [*Lochbühler et al.*; 2013] as shown in Figure 9. For our work we will focus on the well pair P2-P3, where P3 is the source well and P2 is the observation well, some 3.5 m to the west. The tomographic system consists of two double-packers in each well, where the extent of the isolated regions was 0.25 m and the spacing of the intervals was 0.5 m. A suite of observed pressure variations for receivers in the observation well are shown in Figure 10. We will be interested in the propagation time of the pulse, as measured by the arrival time of the peak pressure at each observation point, which is referenced to the time at which the peak pressure is obtained in the source interval.

The overall inversion methodology was discussed and illustrated above and the details will not be repeated here. During one step of the iterative linearized algorithm we minimize the weighted sum of the squared misfit, the model norm, and the model

roughness, as given in equation (24). The requisite equations are given by the conditions that the total misfit is minimized, that is by the equations that result from setting $\nabla \Pi$ equal to zero, where the gradient is taken with respect to the components of $\delta \mathbf{s}$. Thus, at each iteration we solve the set of linear equations (25) for the perturbations in $\mathbf{s}$. The misfits $\Pi(\delta \mathbf{s})$, plotted as a function of the number of updating steps in the iterative inversion algorithms, are shown in Figure 11. The eikonal equation residuals, calculated by the reservoir simulator, tend to level off after about 3 iterations and decrease gradually as the inversion algorithm progresses. This may reflect the fact that as the heterogeneity increases, the eikonal paths begin to deviate from the actual trajectories, as illustrated in *Vasco* [2018]. The match to the observations is shown in Figure 12 for both the eikonal-based inversion and the inversion based upon the extended trajectories. The error reduction of 76% for the extended inversion, shown in Figure 11, is generally monotonic. The error reduction for the extended solution is significantly larger than that for the eikonal-based inversion. Both algorithms improve the fit to the observed arrival times though considerable scatter remains in the residuals (Figure 12).

The final models produced by the two inversion algorithms are plotted in Figure 13. Both models display generally higher permeabilities at shallower depths with values decreasing as the lower edge of the model is approached. The anomalies are largely horizontal, suggesting a generally layered structure, in agreement with previous studies [*Klotzsche et al.*, 2010; *Lochbühler et al.*, 2013; Jimenez et al., 2016; *Somogyvari et al.*, 2017; *Kong et al.*, 2018]. The magnitude of the permeability variations is larger in the trajectory mechanics-based inversion and a high permeability layer is evident in Figure 13. These general features are observable in the upper and lower permeability bounds plotted as a function of elevation in Figure 14. Both models display a decrease in permeability with depth, but the variations in the eikonal-based inversion are somewhat smaller than those of the extended trajectory approach.

We can compare our results to previous work by *Lochbühler et al.* [2013], where a joint inversion of crosswell ground-penetrating radar traveltimes and hydraulic tomography (travel times and amplitudes) was discussed. In Figure 15 the spatial variations of the logarithm of hydraulic conductivity corresponding to our inversion grid are plotted to the same color scale. These results correspond to part of Figure 4h in *Lochbühler et al.* [2013]. In addition, we extracted the highest and lowest permeability values as a function of depth in the inversion region and the average permeability at each elevation. All results show the same general decrease of permeability with depth in the aquifer, as the clay aquitard is approached. The variations in permeability in the extended approach are of the same order as the joint inversion result. As in the synthetic case, the magnitude of the variations in the eikonal equation inversion is smaller.

As a validation effort, we left out data from the fifth source from the bottom in Figure 13 when conducting the inversion for the permeability multipliers. This allowed us to use the resulting models of $K$ variation shown in Figure 13 to estimate the travel times of pressure pulses from the source at position five to the corresponding observation points. The resulting observed and calculated travel times from this experiment can then be used to validate the model as indicated in Figure 16. There is considerable scatter in the arrival times but the overall trend is a variation that increases in correspondence with the observed arrival times. The largest disagreement is between an eikonal-based arrival time estimate and the observed value, but the overall scatter seems comparable for the eikonal and extended methods. The largest deviations are the large predicted travel times for the arrivals observed at around 1.35 s. Such long travel times might be due to the significant low permeability values near

source position five. The source-receiver distribution is rather sparse, and there may not be sufficient redundancy to conduct an accurate validation experiment.

Lastly, we have used equations (28) and (29) to calculate the diagonal elements of the matrices $\mathbf{R}$ and $\mathbf{C}_{ss}$, respectively, in order to assess our trajectory-based solution. The diagonal elements of the matrices are plotted in Figure 17, in the locations of the grid blocks that they represent. The diagonal coefficients of the resolution matrix are close to 1 if the parameter can be determined without interference from other grid block estimates [*Vasco et al.*, 1997]. That is the i-th row of the resolution matrix are averaging coefficients associated with the i-th parameter. The row approaches a delta-function like distribution and the diagonal element approaches the value 1 when there is little averaging with other parameters. The diagonal elements of the resolution matrix in Figure 17, with peak values around 0.6, indicate moderate spatial averaging in these estimates. In particular, there is greater averaging than in the synthetic test due to the fact that we are only using sources situated in a single well in the field case. The spatial averaging is greatest and the resolution poorest for the grid blocks at the edges of the model, particular at the top of the crosswell region. Similarly, the model errors, also shown in Figure 17, are larger than in the synthetic test, around 20% of the size of the model estimates. The resolution and covariance estimates indicate that the high permeability layer, located in the upper portion of the model, is moderately well constrained by the observations. Due to sampling issues the error estimates are not reliable at the edges of the model and tend to zero where there are few or no trajectories. As indicated in the synthetic test, putting sources in both wells would increase the resolution and reduce the uncertainty associated with our estimates, suggesting how we might improve our imaging in the future.

## 4  Conclusions

The trajectory mechanics approach described in *Vasco* [2018] and applied here is very general and can be used to model other hydrological processes such as tracer transport [*Vasco et al.*, 2018] and multiphase fluid flow. One advantage associated with transient pressure is the rapid propagation of a disturbance in comparison with the velocities associated with fluid transport. Thus, transient crosswell pressure testing can be conducted relatively rapidly in formations with moderate hydraulic conductivity. This is particularly true when transient pressure travel times, such as the arrival time of the peak of a pressure pulse or the peak of the time derivative of the pressure [*Vasco et al.*, 2000] are used. For the Widen field experiment the peaks are observed in the first few seconds of the measured traces in the adjacent borehole. Another advantage of hydraulic travel time tomography is that the relationship between the arrival times and the hydraulic conductivity or diffusivity is quasi-linear [*Cheng et al.*, 2005]. Thus, the final model resulting from an inversion of travel times is less sensitive to the initial or starting aquifer model and less likely to become trapped in a local minimum. Finally, travel time tomography provides an element of data reduction, from an entire transient pressure waveform, to a single arrival time. This can be advantageous when dealing with many intervals from multiple boreholes, time-lapse pressure changes, or large data sets derived from geophysical observations.

We have presented two examples of hydraulic tomographic imaging, one using synthetic transient pressure arrival times and the other using data from an experiment at the Widen field site on the Thur River in northern Switzerland. We do find that an algorithm based upon the eikonal equation is significantly faster than one utilizing the extended trajectories calculated

using a reservoir simulator, taking only about 10 seconds compared to 129 minutes. From the synthetic application we find that an imaging technique based upon the eikonal equation, the current method used for trajectory-based modeling, has difficulty accurately imaging large and abrupt changes in permeability. Such rapid spatial changes in flow properties are a common occurence in geologic media, with the presence of layering and fractures, with correspondingly large variations in hydraulic conductivity. For example, in well logs it is quite common to observe thin layers with permeabilities that are orders of magnitude larger than values in the surrounding formations. Indeed, in our field case at the Widen field site we image an order of magnitude change in permeability in agreement with previous results at the site. While the eikonal equation is much faster and can recover large-scale spatial variations, it is likely to produce smoothed images of sharp features and to underestimate rapid changes in properties. Thus, the approach is useful as a rapid reconnaissance tool, as in real-time imaging, and for regions where the properties are thought to be smoothly-varying. This usage is supported by that fact that both the eikonal-based and the extended trajectory-based methods share the quasi-linearity of travel time inversion approaches [*Cheng et al.*, 2005], and are less sensitive, in comparison to inversions based upon head magnitudes, to the initial or starting model.

For a full analysis and interpretation of field data however, we recommend the trajectory mechanics approach, as it does not invoke assumptions about model smoothness and is therefore more robust and accurate, yet it retains the semi-analytic sensitivities that are characteristic of trajectory-based approaches. The semi-analytic sensitivities are computed after a single simulation, using either numerical methods to solve the coupled system for $\mathbf{x}$ and $\mathbf{p}$ or using a numerical simulator to determine $\mathbf{p}$ directly. Even if one resorts to a numerical simulation, the semi-analytical nature of the sensitivities provide some advantages over conventional methods. The most efficient conventional method for computing numerical sensitivities is based upon adjoint methods and requires the formulation and solution of the adjoint equation along with an additional simulation to calculate the residuals. Thus, two simulations are required in order to estimate the sensitivities for a given test.

The approach that we have described is useful for imaging permeability variations between boreholes but it does have some limitations. The use of slug tests limits the allowable distance between wells that may be used for imaging variations in $K$. However, as noted in *Vasco et al.* [2000], one can use a constant rate test and consider the arrival time of the steepest slope, extending the range of the test to larger offsets between wells. We have chosen to fix the reservoir storage and determine variations in an effective $K$. This assumption needs to be explored in future studies and tested under realistic conditions. The computation aspects of this approach are significant, requiring full reservoir simulations for the inversion. As noted in *Vasco* [2018], there are more efficient methods that involve solving the equations for the trajectory directly, without a reservoir simulation. This should reduce the computation burden of the approach at the cost of a more complicated implementation.

Z

*Acknowledgements.* Work performed at Lawrence Berkeley National Laboratory was supported by the US Department of Energy under contract number DE-AC02-05- CH11231, Office of Basic Energy Sciences of the US Department of Energy. The pressure data from the Widen field test is available on the Zenodo archive. The reference for this data is Vasco, Donald, Doetsch, & Brauchler. (2018) Widen Field Test Pressure data - P02 Experiment. Zenodo.http:// doi.org/10.5281/zenodo.1445756.

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

Governing Equation: $\nabla \cdot (\mathbf{K} \cdot \nabla h) = \zeta \dfrac{\partial h}{\partial t}$

Substitute
$h(\mathbf{x}, t) = e^{-S(\mathbf{x},t)}$

Ray Equations:

$$\frac{d\mathbf{x}}{dt} = \frac{1}{\zeta} \mathbf{p} \cdot \mathbf{K}$$

$$\frac{d\mathbf{p}}{dt} = \nabla \left[ \frac{1}{\zeta} \nabla \cdot (\mathbf{K} \cdot \mathbf{p}) \right]$$

Solve
Numerically

Use
Simulator

$$T = \int_{\mathbf{x}} \frac{1}{v} dx = \int_{x} s\, dx \qquad \mathbf{p} = -\frac{\nabla h}{h}$$

Perturbation Method

Sensitivity $\qquad \delta T = \displaystyle\int_{x} \delta s(x)\, dx$

**Figure 1.** Schematic illustration of the approach used to obtain the sensitivities that form the basis for the linearized, iterative, crosswell imaging algorithm.

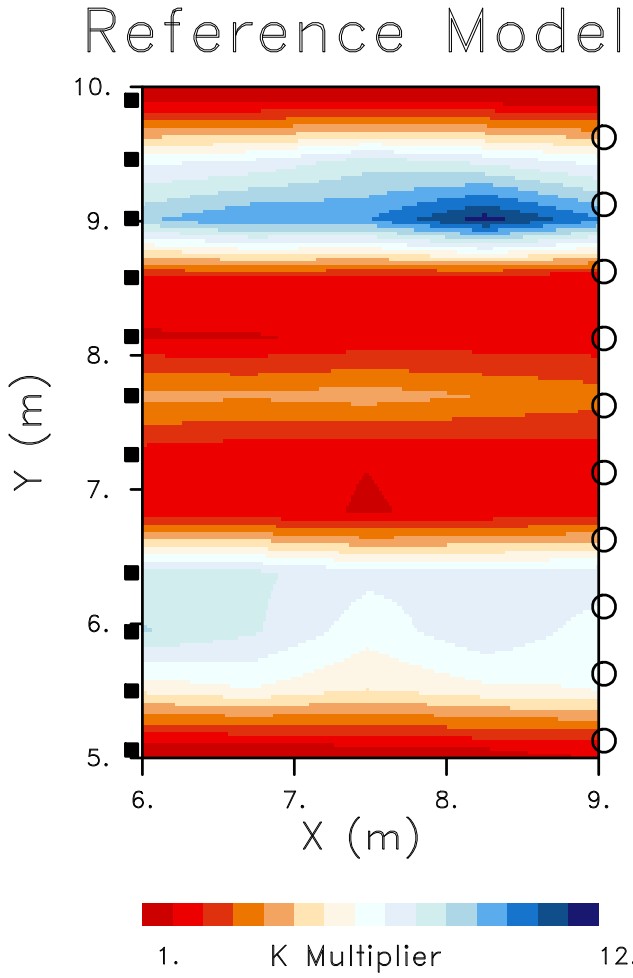

**Figure 2.** Reference model for the crosswell test example. A cross-section through the permeability model representing the crosswell plane. The crosswell configuration, for imaging flow properties between two boreholes, consists of pressure sources in the two wells (filled squares and open circles) transmitting transient pulses to receivers (open circles) in an adjacent well. The source-receiver geometry mimics that of the field experiment conducted in Widen, Switzerland. The color scale varies linearly between permeability multipliers from 1 to 12.

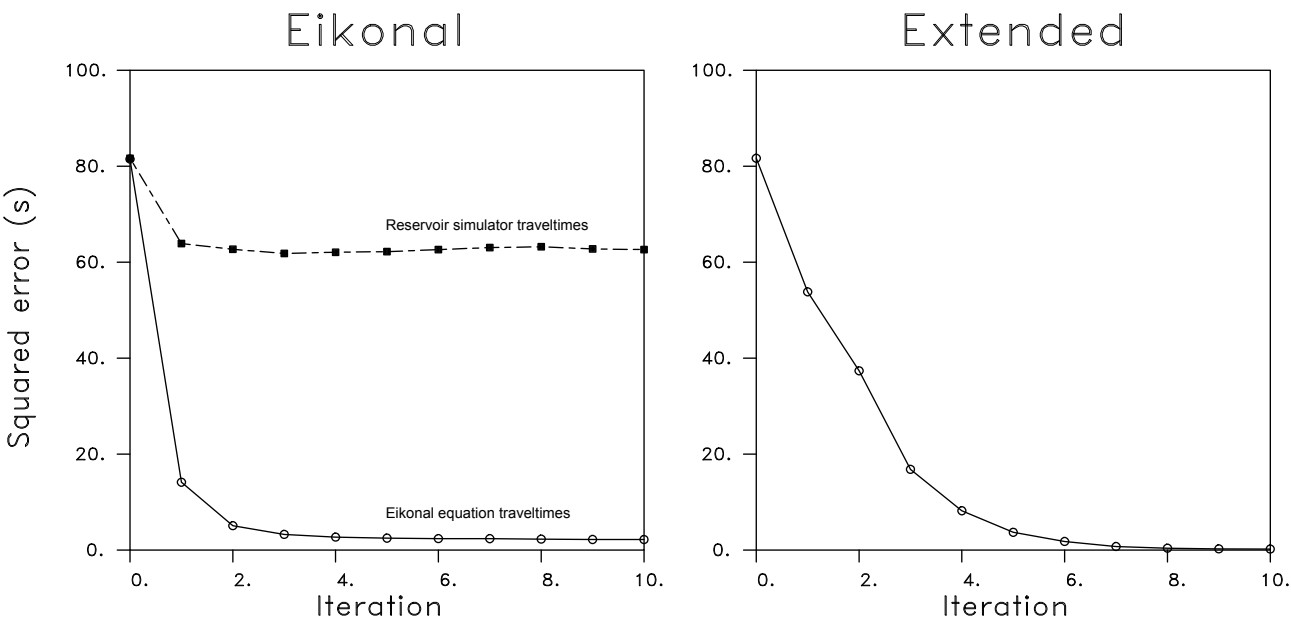

**Figure 3.** The sum of the squares of the residuals for the eikonal-based (Left panel) and extended inversion (Right panel) algorithms as a function of the number of steps in the iterative updating algorithm. For the eikonal equation-based approach two sets of errors are displayed, those produced by the eikonal equation and those produced by the reservoir simulator. The reservoir simulator errors result when the current permeability model is used in conjunction with the TOUGH2 simulation code [*Pruess et al.*, 1999] to calculate head variations and arrival times at the receiver locations.

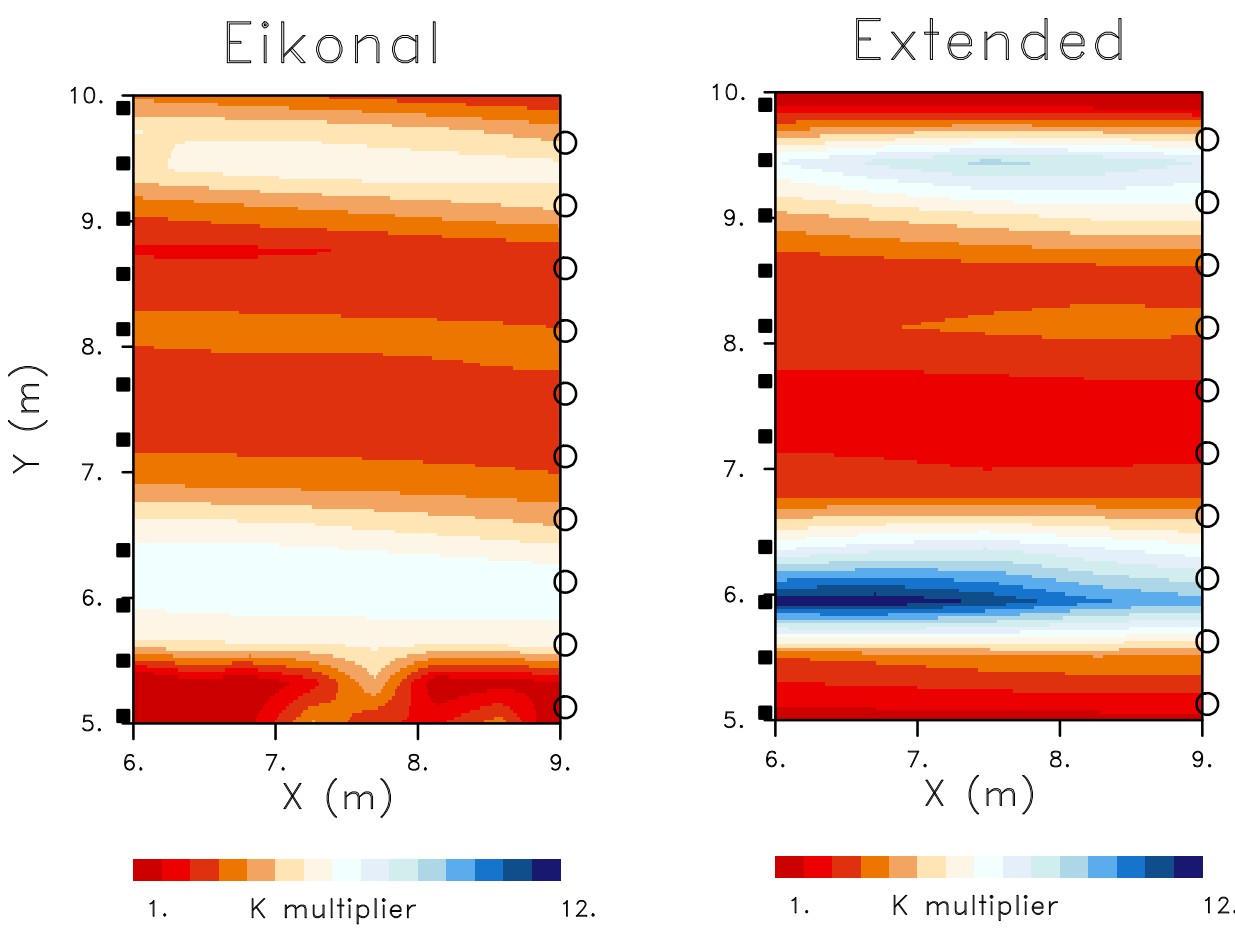

**Figure 4.** The spatial variation in the permeability multiplier resulting from inversions based upon the eikonal (Left panel) equation and on the extended trajectory-mechanics (Right panel) algorithms. The color scale varies linearly between permeability multipliers from 1 to 12.

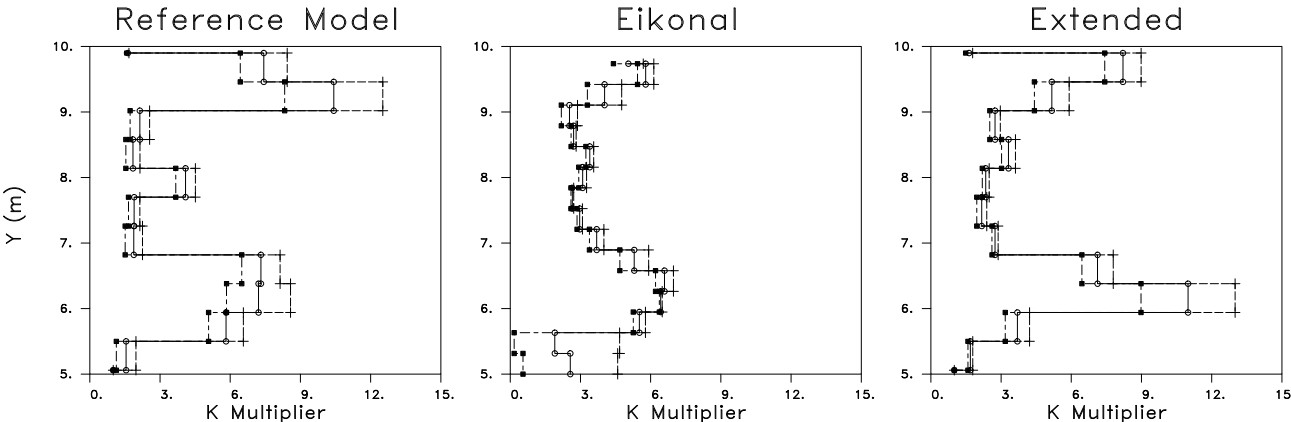

**Figure 5.** (Left panel) Vertical variation of the $K$ multiplier for the reference model, the eikonal-based inversion, and the extended trajectory-based inversion algorithm (solid lines). The maximum and minimum values of $K$ and each depth interval are also plotted in each panel. Only the variations within the crosswell plane, from 5.0 to 10.0 m in elevation are shown.

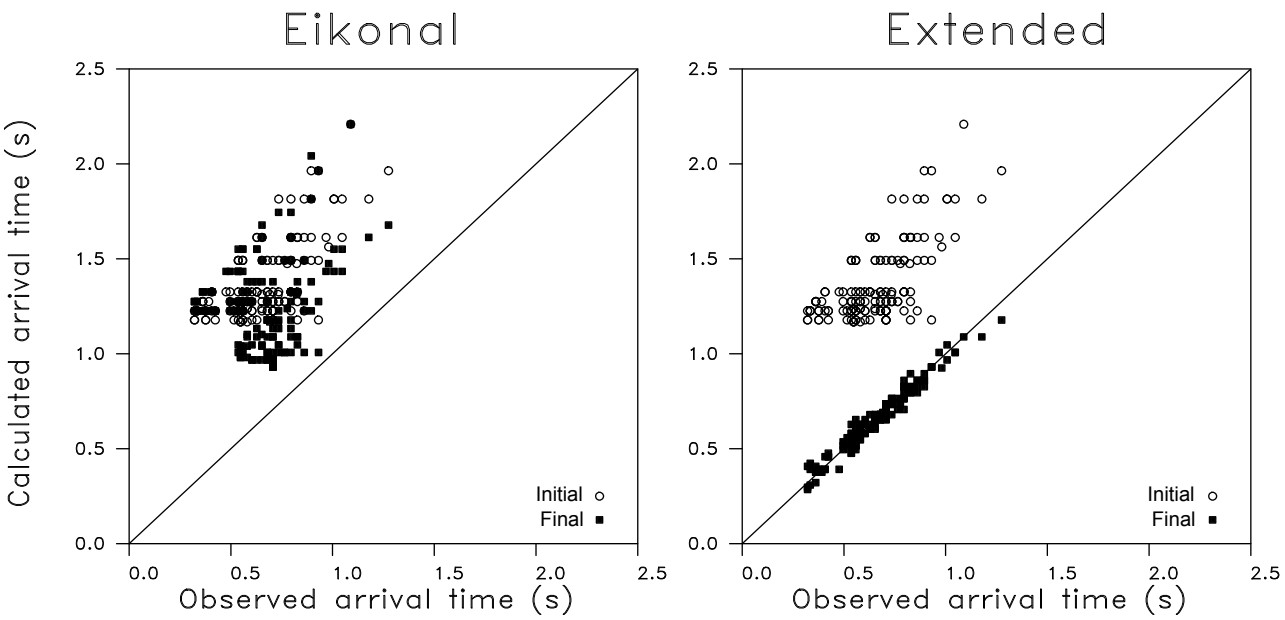

**Figure 6.** Observed versus calculated arrival times for both the eikonal-based (Left panel) and extended trajectory-based (Right panel) inversion algorithms. The initial travel times, calculated using the uniform starting model, are plotted as open circles. The travel times calculated using the final model of each approach are plotted as filled squares.

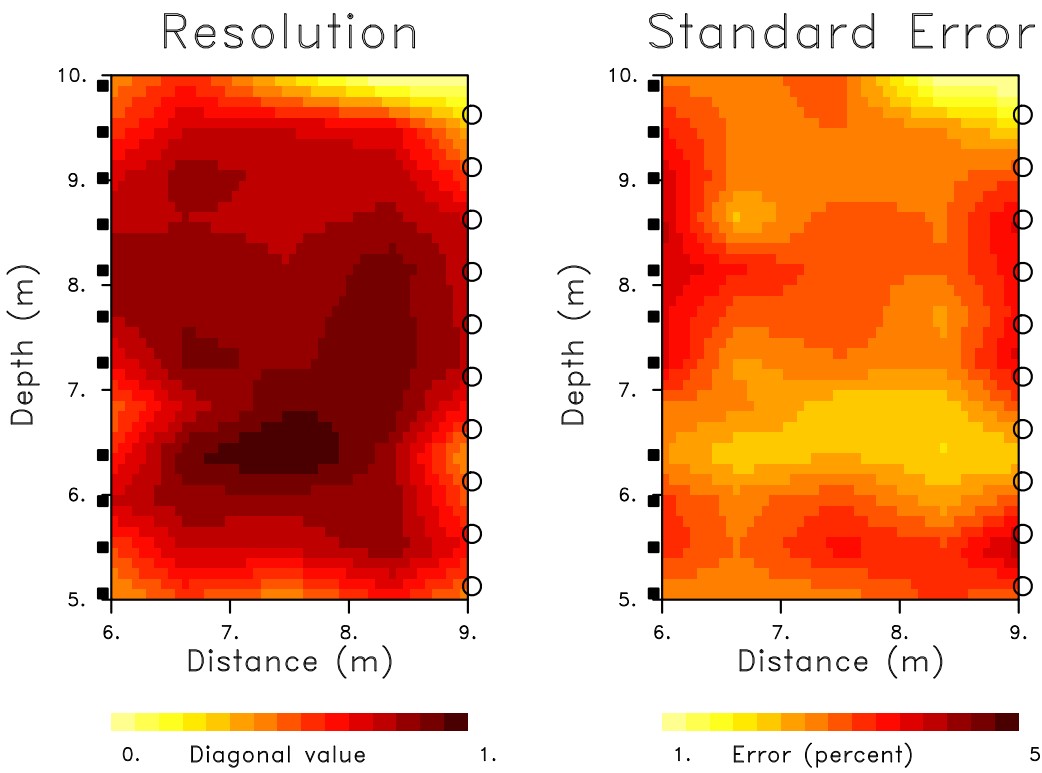

**Figure 7.** (Left panel) Diagonal elements of the resolution matrix indicating the ability to determine the value of a parameter independently of surrounding parameters. Values near 1.0 indicate a well resolved propertie that does not trade-off with values in adjacent grid blocks. (Right panel) Model parameter standard errors as a percentage of the average model update.

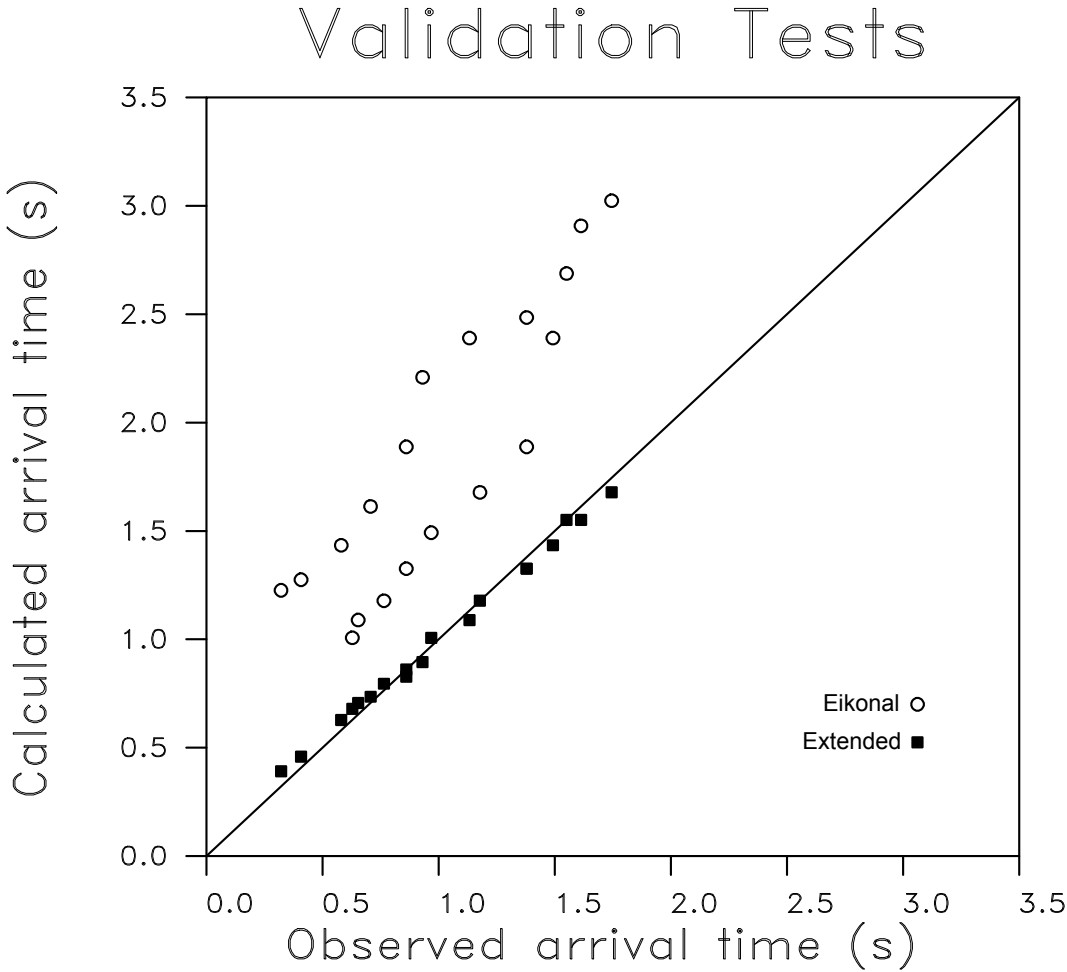

**Figure 8.** Validation exercise in which arrival times for two tests that were not used in the inversion are calculated based upon the final models estimated using the eikonal and extended approaches. These calculated times are plotted against traveltimes computed using the reference model.

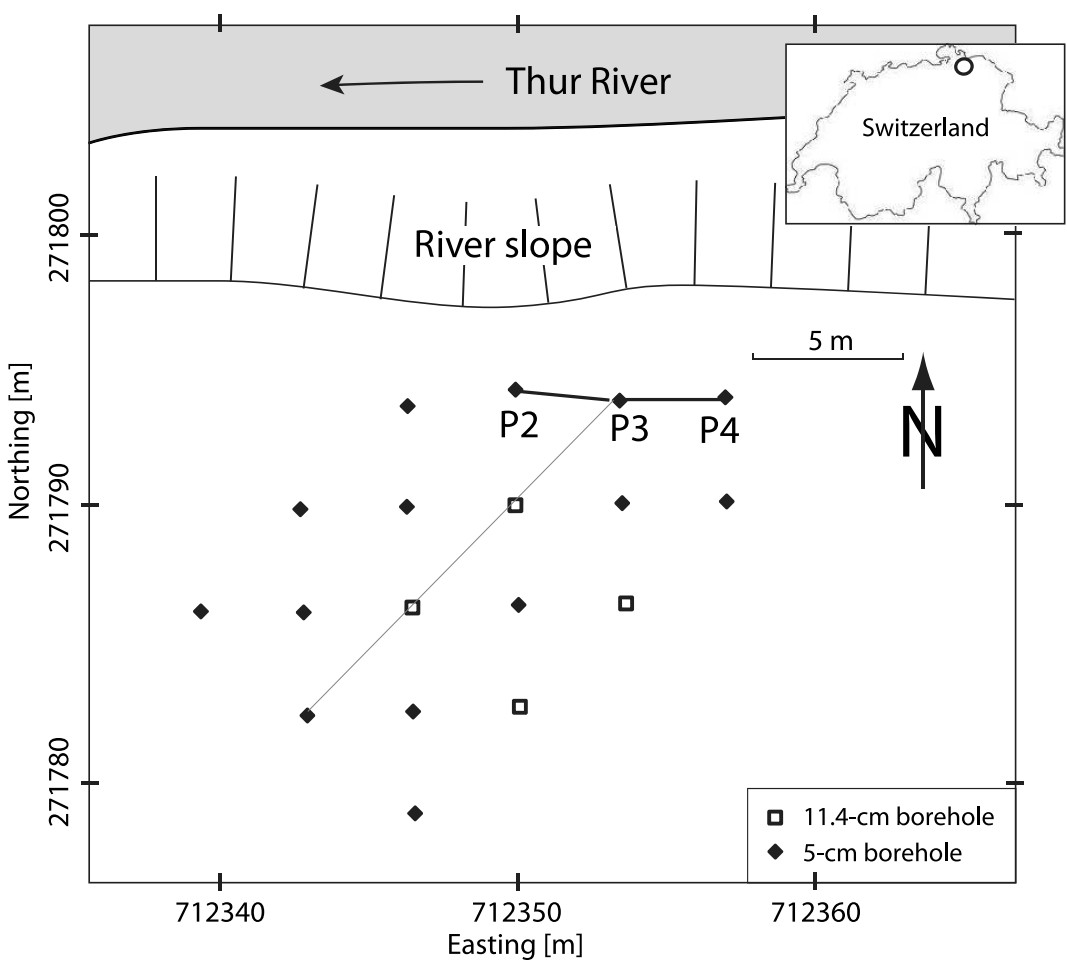

**Figure 9.** Schematic map of the Widen field site located adjacent to the Thur River in Switzerland, as indicated in the insert. The labeled wells P2, P3, and P4, were used for several hydraulic tomographic experiments.

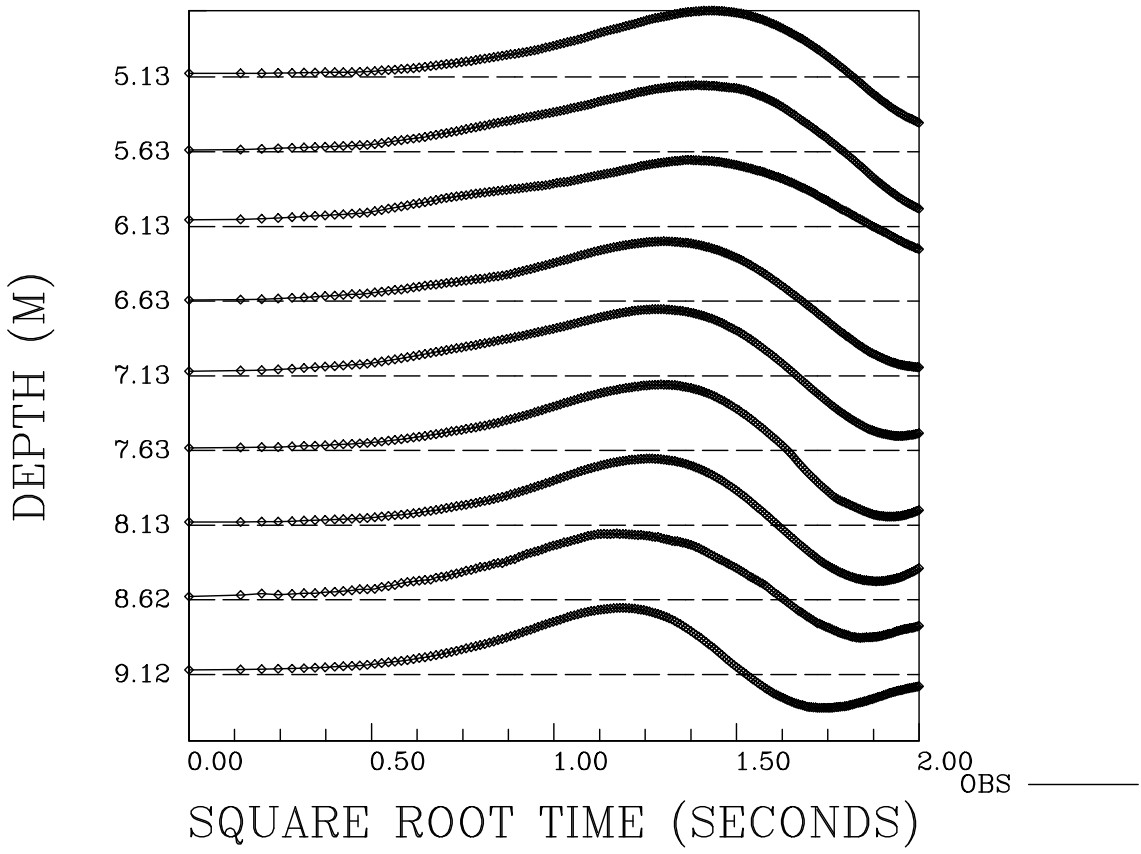

**Figure 10.** Hydraulic head, from a crosswell slug test, recorded at a set of packed-off intervals in observation well P-2 from the Widen field site. Each trace has been normalized in order to have a unit peak amplitude.

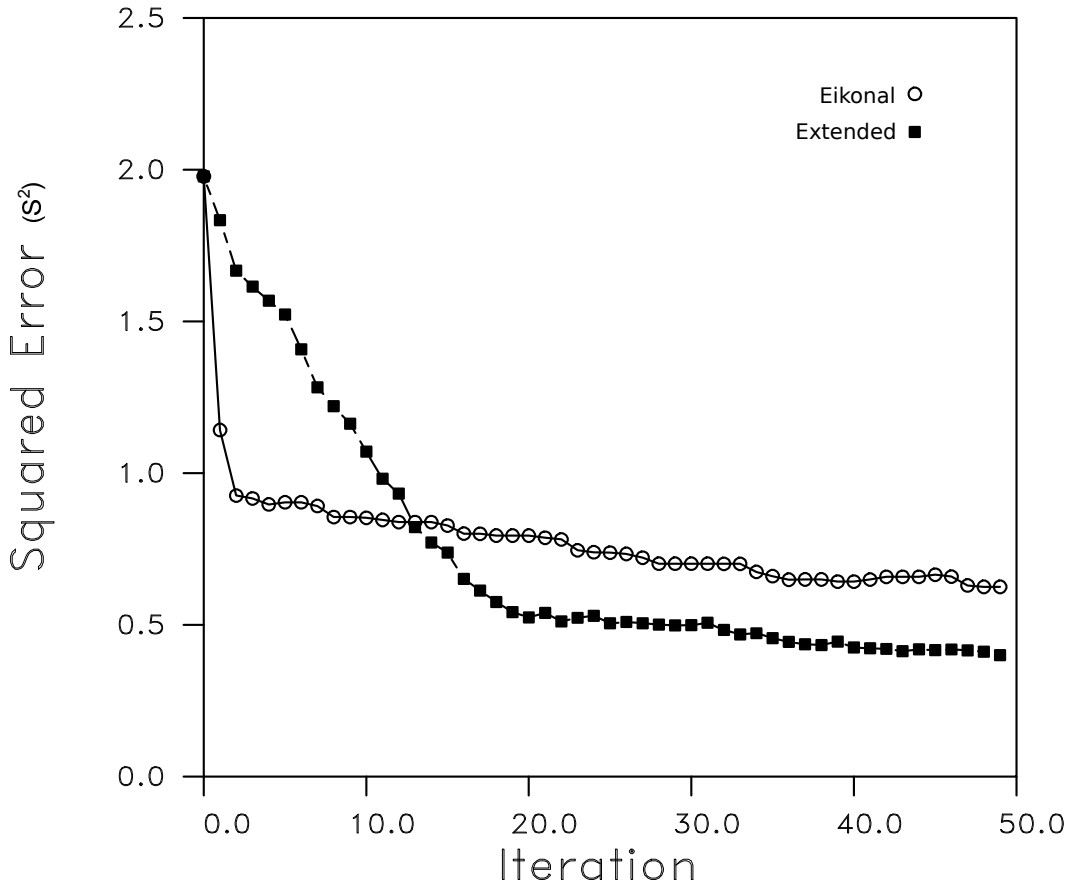

**Figure 11.** Mean squared error for an inversion of the hydraulic head arrival times. The inversion labeled eikonal is based upon the eikonal equation and uses high frequency asymptotic trajectories. The open circles are the calculated mean squared error calculated using travel times produced by the numerical simulator TOUGH2. The filled squares denote the mean squared error as a function of the number of iterations of an inversion scheme that utilizes the extended trajectories that follow from equation (9).

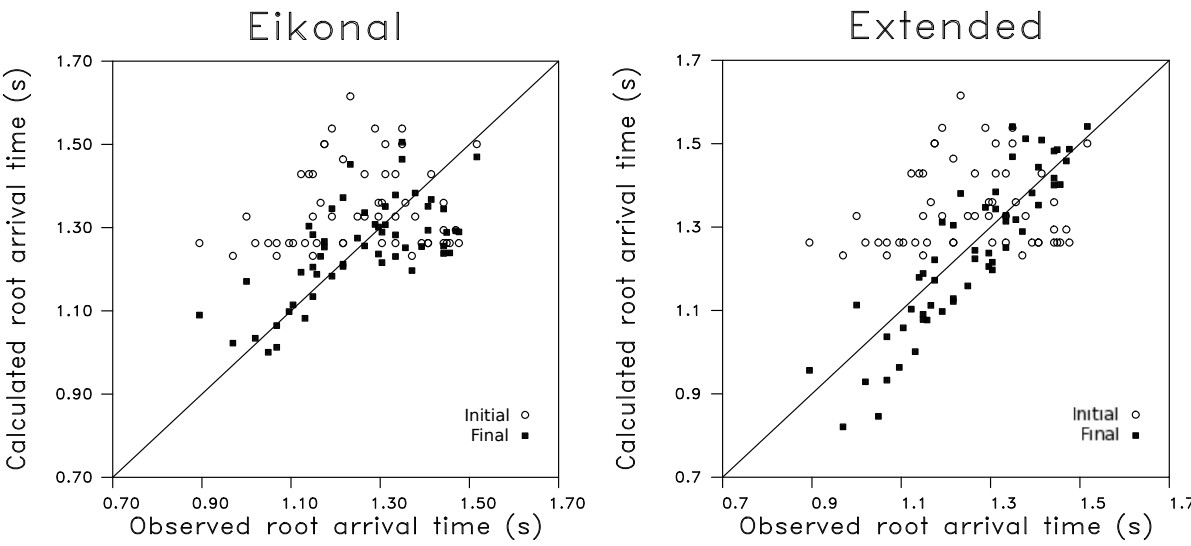

**Figure 12.** Initial (open circles) and final (filled squares) misfits for both the eikonal-based and trajectory mechanics-based inversions. (Left panel) Calculated arrivals, based upon the numerical simulator TOUGH2, run with the models from the eikonal-based inversion. The calculated arrivals are plotted against the observed arrivals, for a perfect match the points would lie along the diagonal line. (Right panel) Calculated travel times plotted against the observed arrival times for the inversion that uses the extended trajectories that result from solving equation (8).

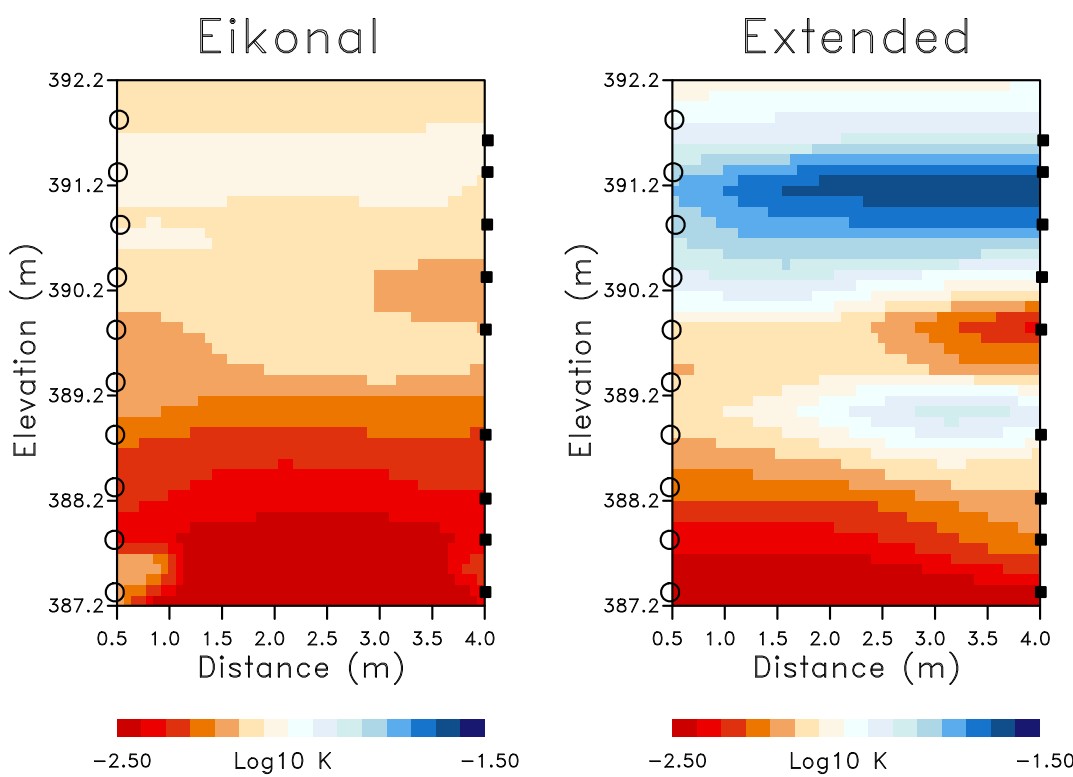

**Figure 13.** (Left panel) Permeability multiplier estimates produced by the iterative updating algorithm based upon high frequency asymptotic trajectories. (Right panel) Estimates of permeability multipliers resulting from an iterative inversion method that is based on the extended trajectories calculated using equation (8).

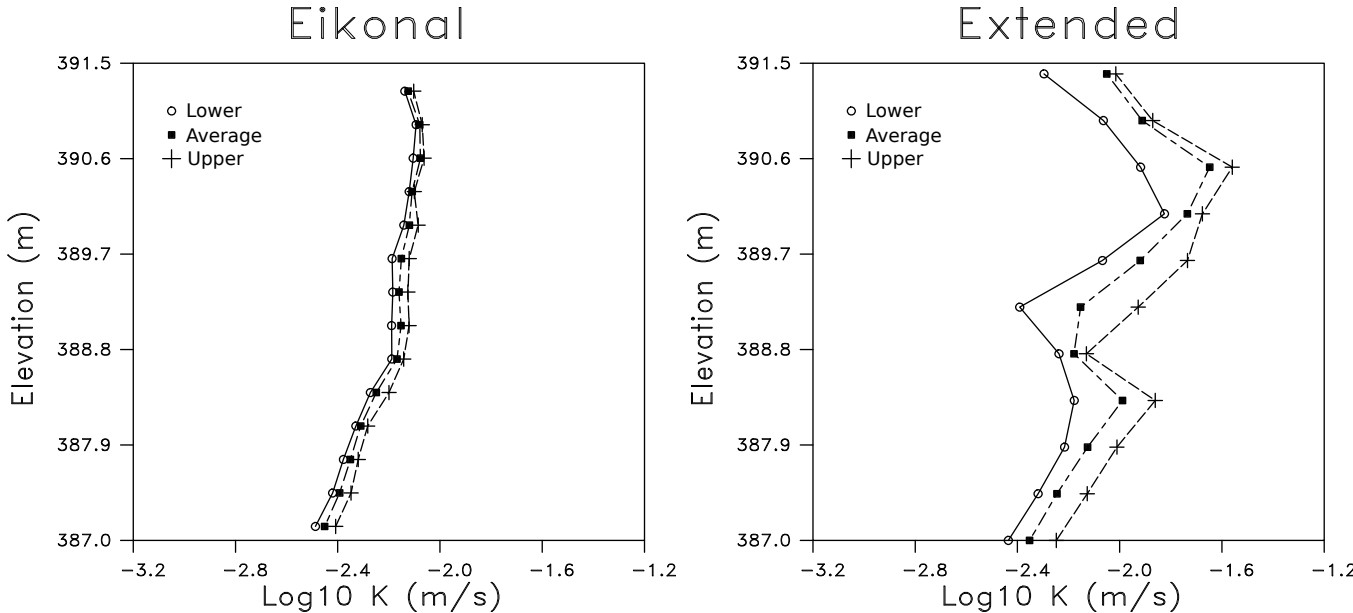

**Figure 14.** Upper (crosses) and lower (open circles) permeability values as a function of elevation within the model. The laterally averaged permeabilities are also plotted as filled squares in each panel.

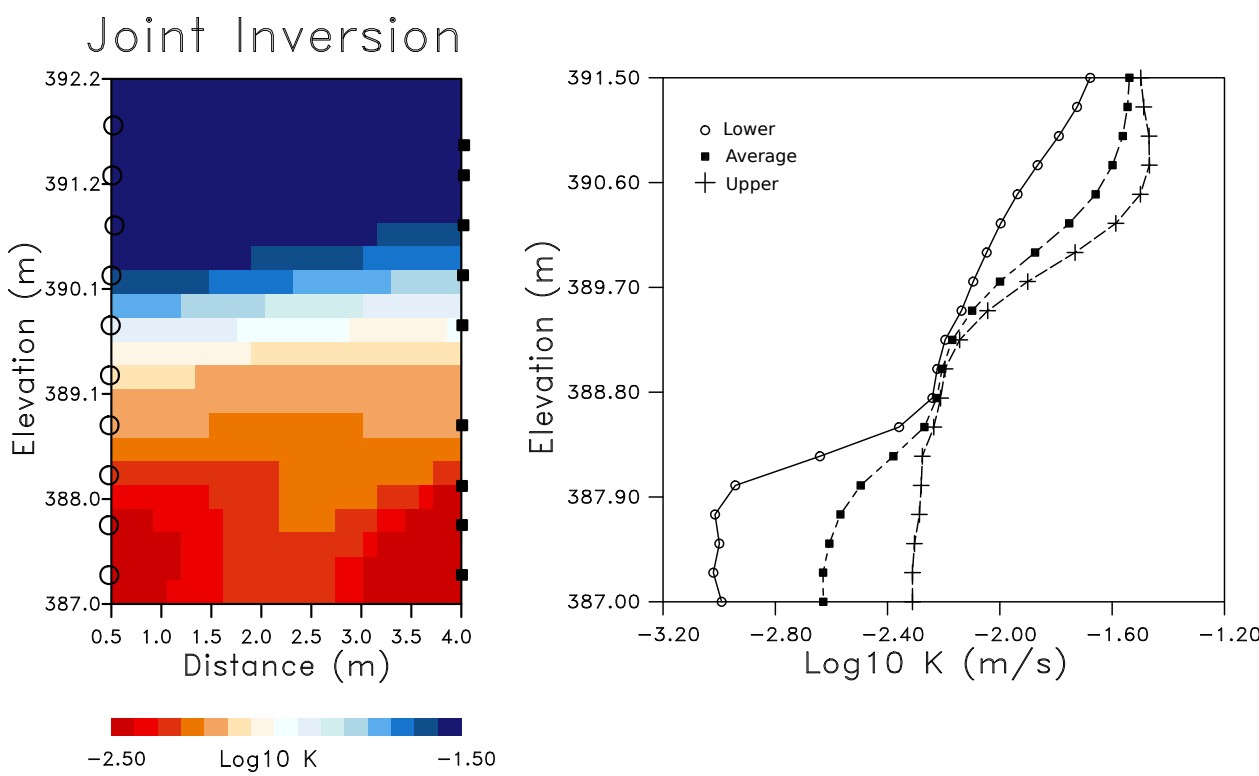

**Figure 15.** (Left panel) The portion of the inversion results of *Lochbühler et al.* [2013] that corresponds to our inversion domain. Their joint inversion includes ground-penetrating radar travel times as well as travel times and amplitudes from crosswell slug tests. (Right panel) The highest and lowest permeabilities at each depth in the inversion domain, plotted along with the average permeability as a function of elevation.

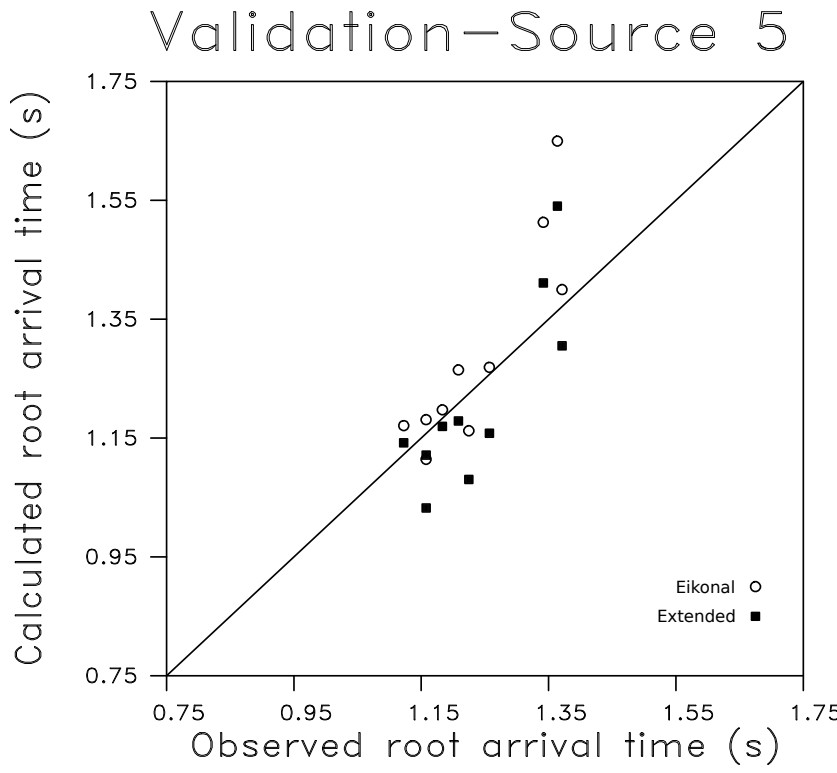

**Figure 16.** Validation test in which arrival times from source 5, which was not used in the inversion, are calculated based upon the final models estimated using the eikonal and extended approaches. These calculated times are plotted against the observed traveltimes.

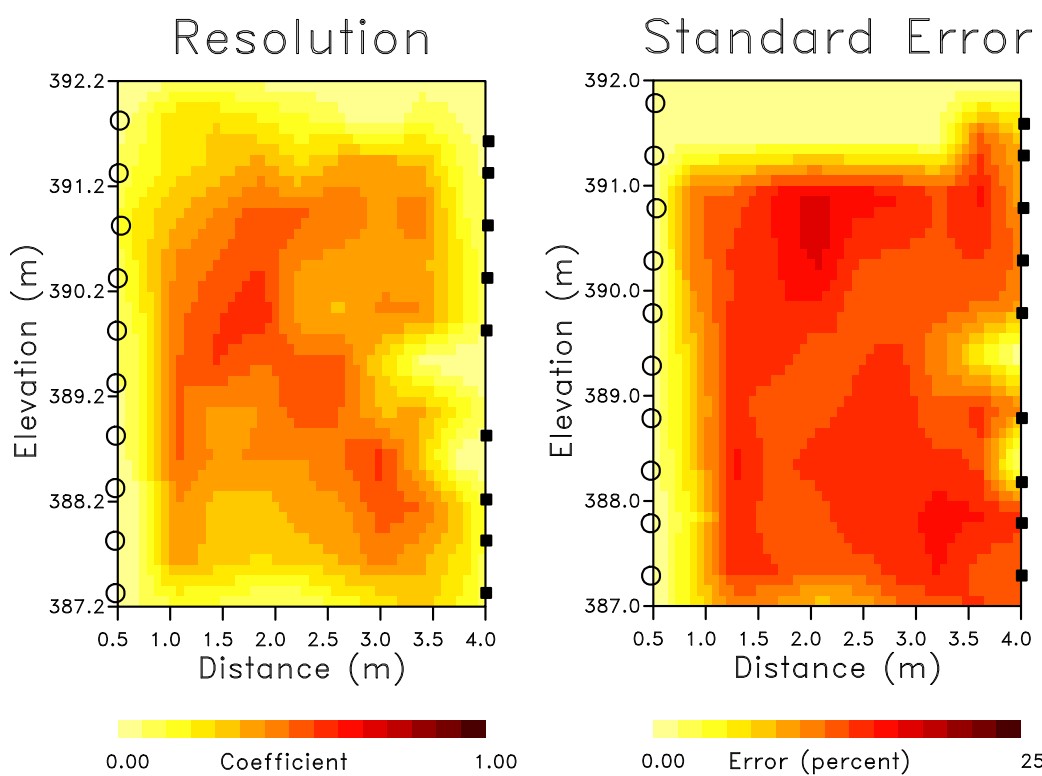

**Figure 17.** (Left panel) Diagonal elements of the model parameter resolution matrix associated with the Widen field experiment. Diagonal elements with values near 1.0 are well resolved, that their estimates do not trade-off with the values of other parameters. Values near zero are not well determined. (Right panel) Square roots of the diagonal elements of the covariance matrix plotted in the location of the corresponding grid block.