# Peer review of "An extended trajectory-mechanics approach for calculating the path of a pressure transient: Traveltime tomography"

_Hydrology and Earth System Sciences, 2019_

## Referee Comment (RC1) · Anonymous Referee #1 · 16 Jun 2019

Note to editor: I posted my comments in plain text mode, but the formatting is lost. I requested Editorial for suggestions on how to send PDF/WORD files while maintaining anonymity. I can send these files directly to you if needed.

_______________________________________________ There are several issues to consider when assessing an inverse modeling approach. They must all be addressed in order for an inverse modeling approach to be considered as a viable option in application. I believe this paper addresses only a subset of issues, and additional work is required prior to publication. I will address these issues in the form of Q&A. The title of the paper is confusing, as it puts an emphasis on the solution to the

Eikonal equation. Now, if this is indeed to goal, then the paper should be resubmitted to any math journal. But as stated in the abstract, the paper attempts to present an inverse modeling/subsurface characterization approach. As such, it must address a wide range of issues (which, to a large degree are ignored), and this is its main weakness. Here are several issues to consider: 1. Is the math solid? I did not check the math, but the authors have been at that for many years and the results look reasonable. So I trust it to be OK, but it is neither relevant nor interesting for HESS readership. 2. Does this paper present a complete approach for inverse modeling? a. Uncertainty Quantification (UQ): UQ is ignored. Uncertainty quantification is not mentioned, let alone addressed. Tons of ink was spilled in writing on the significance of UQ, which I won't repeat here. b. Regression/optimization: Does an image produced following an optimization-based approach provide a realistic representation of reality? Not even by a long shot. Especially when UQ is ignored. c. Data integration: Field surveys usually provide multiple types of data. An inverse modeling approach MUST provide a way for integrating all types of information into a coherent image of the subsurface. We do not have this feature in the proposed approach. This would be acceptable for non-invasive surveys, but not for tomographic surveys. d. Spatial coverage is very limited: The survey boreholes (doublet) are very close to each other, thus covering a small volumetric fraction of regular size domains under investigation. This implied the need to develop multiple doublets, which is difficult to imagine. How far apart can the boreholes be? And when you have multiple wells, is it reasonable to ignore core data? See 2(c). e. Use of penalty terms and linearization: See discussion following equation 23. How do they affect the solution? What are the implications in terms of application? Which type of formations respond well to these math manipulations? f. Conclusion: The paper fails to present an acceptable inverse modeling strategy 3. Is the case study presented appropriate? And what can we learn from it? a. The survey boreholes are very close to each other (2-3 meters). I can imagine that when drilling a borehole, one would obtain a few cores and testing in the lab. With the wells being so close to each other, one can contemplate some sort of interpolation (e.g., geostatistics-based) of the core data.

Question is then, would tomography do better? This must be shown. What I would to see is a comparison between the following 3 options: (1) core data only, (2) geophysics only, and (3) combination. b. The authors provide a 2D solution (they solve the Eikonal equation in a vertical plan) to a 3D problem, as there is no guarantee that nature would follow the 2D geometry. It would be important to evaluate how good such (2D, planar) an approximation is: what type of formations are amenable for such treatment, or not? What's the ideal and max spacings between the boreholes in real-life applications? If the type of geology is not an issue, this should be shown. And that brings us to the following point: c. The authors chose a layer-cake formation for their case study, which is favorable to the 2D approximation. We need to see an application that does not play so favorably into this 2D/3D issue. d. Following on item 3c, the second line in the abstracts claims applicability to "general porous media". What does "general" mean? And what is the factual basis for this statement? On p. 2 it is stated that the method only applicable to smoothly varying formations, which raises some concerns about accuracy of the "general porous media" statement. The case study does not support the generality statement. e. Conclusion: Case study is not adequate.

---

## Author Comment (AC1) · 21 Jun 2019

While there are things that we do agree with in this review, as a general comment we feel that much of the criticism is not directed at the limitations of the new method, rather the remarks point out limitations on the model assessment and the drawbacks of crosswell slug tests. To answer the issues in order:

It is not clear how the title invokes the Eikonal equation (the term is not mentioned), other than we use a trajectory-based approach for tomography. Simply put, the paper is concerned with a semi-analytic approach to pressure arrival time tomography.

[Figure]

1. Is the math solid? Yes, we have developed trajectory-based imaging approaches for tracer and transient pressure data over many years. However, this paper is concerned with a new approach that overcomes well-known limitations of previous asymptotic methods and should be judged on it's own merits. The work hardly belongs in a math journal, as it's basis comes from developments in physics and the physics of wave propagation (Wyatt 2005).

2. Does this paper present a complete approach for inverse modeling? We are not trying to present a complete approach for inverse modeling. Instead, we present a new approach for working with cross-well slug tests. These tests are quite common and have been shown to yield important additional information that can be combined with core analysis, borehole logging and any other available data.

(a) Uncertainty Quantification. We do discuss model parameter resolution and uncertainty at the bottom of page 8 and provide several references (Vasco et al. 1997; Bohling, 2009; Paradis et al. 2016), noting how the linearized expression (25) facilitates such computations, so the reviewers comments are both unfair and inaccurate. We agree that an important part of the solution to the inverse problem is model assessment and we will revise the paper to include such an assessment. However, as discussed in the current paper and as we will show in our revision, the trajectory-based approach does not preclude the calculation of model parameter resolution and uncertainty but rather facilitates such computations.

(b) Regression/Optimization. Any model is a simplification of reality. However, models are very helpful e.g. for flow and transport modeling, e.g. for prediction of arrival of a tracer or pollutant. The approach does recover the main anomalies present in the reference model, even though they are roughly an order of magnitude larger then the values in the uniform initial or starting model. The resulting model both fits the synthetic pressure arrival times and the validation arrival times. As such, the model derived from the inversion could serve as the starting point for a complete matching of the transient pressure data. (c) Data Integration. The approach that we are describing can easily be

used as part of an integrated inversion scheme in which many data types are included. Nothing in the methodology prevents that. Therefore, to say that the trajectory-based approach does not provide a way to integrate different types of information is simply incorrect. We merely chose to focus solely on the inversion of pressure arrival times in order to keep the presentation simple and direct and not obscure the comparison with an eikonal-based approach for inverting arrival times.

(d) Spatial Coverage. There are limitations in the crosswell geometry and in slug tests for imaging properties between wells. These limitations are well known. But that is not an issue associated with the new approach that we are illustrating. As the approach that we are describing is applicable to any test where a simulator can be used to compute the transient pressure response at an observation point. In fact, as illustrated in Vasco et al. (2000), the travel time approach can be used for wells 100 m apart when a constant rate test is considered. In that case the peak of the slope of the transient pressure curve is used to define an arrival time. In addition, fully three-dimensional imaging is also possible if such well configurations and data are available. The new technique is not limited to any specific geometry, such as the two-dimensional plane dictated by our crosswell experiment. We merely used data that were available to illustrate the approach. Given three-dimensional crosswell constraints provided by a suite of wells, we could apply the method in three-dimensions.

(e) The use of penalty terms and linearization. We can include some discussion on the influence of the regularization. We use an iterative approach to solve the non-linear inverse problem involving incremental linearized updates. We do not linearize the problem. This is a very common approach for solving non-linear problems in geophysics and hydrology and one of the few practical approaches for very large problems.

(f) Conclusion. The goal of the paper was to compare a new trajectory-based technique for pressure travel time tomography to a conventional approach. This conventional approach, based upon transient pressure travel times has provided the basis for many 'acceptable inverse modeling' studies that we cite in the Introduction. We believe that

we have shown that the new approach is somewhat better at imaging rapid variations in properties, as the theory predicts.

3. Is the case study presented appropriate? And what can we learn from it?

(a) Again, we chose to illustrate the approach with data that we had already collected, that was fairly well understood, and that had already been analyzed. The wells are closely spaced as the tests were designed to understand river hydrology. Slug tests, which propagate a transient pulse, cannot be spaced too far due to the decay of the pulse with distance. We do provide a comparison between the new approach using just transient pressure arrival times and an integrated approach that combines geophysical data and pressure data. In the field application the new approach does seem to recover larger variations in permeability (Figure 11), variations that are of the same order as those obtained by a more comprehensive joint inversion (Figure 13). Furthermore, the new approach does seem to have more detailed structure than does the joint inversion. The amount of well core data was limited, and it can be difficult to extract undisturbed core samples from soft sediment environments. However, there are some cores and we can include these observations in a revision.

(b) Crosswell tests in general are limited to imaging within the plane between the two wells. This is just a characteristic of crosswell imaging. One can and does model the pressure propagation in three-dimensions but cannot resolve structure that is far outside of the plane between the wells.

(c) Crosswell tests are generally conducted in regions where the structure is thought to be largely planar with relatively smooth lateral variations and possibly rapid depth variations, is in the field area considered in this paper. The tests were conducted in a sedimentary environment and the wells are close. As such, it is thought that the most rapid spatial changes would be vertical variations in properties. As noted above, nothing in the approach precludes its application in a fully three-dimensional setting, with data from multiple wells. We hope to conduct such studies in the future as we

acquire more complete three-dimensional data sets. However, the goals of this paper were limited and reflect the characteristics of our field observations.

(d) By general porous media we mean any porous medium that can be modeled using a numerical reservoir simulator. The factual basis for the statement is that the computations only rely on quantities output by the simulators, it does not invoke any additional assumptions about the medium. On page 2 it is stated that 'previous trajectory-based formulations...relied upon an asymptotic approach that assumes smoothly-varying properties'. Furthermore, we go on to state 'Here we apply a newly developed trajectory-based technique for travel time tomography that dispenses with the assumption of smoothly-varying properties, enlarging its range of validity to any model that may be treated with a numerical simulator.' I am not sure that I understand how the referee reached their conclusions, the statements seem clear.

(e) To simply say 'The case study is not adequate', is not constructive and provides no guidance on possible improvements.

---

## Referee Comment (RC3) · Anonymous Referee #2 · 23 Jun 2019

The manuscript introduces a new asymptotic algorithm for the inversion of hydraulic tomography data. The methodology improves the tomographic reconstruction compared to the existing eikonal based inversion and provides more accurate hydraulic conductivity reconstructions. This is demonstrated on both synthetic and field examples. The results are validated with independent data. I find the manuscript very well written and easy to follow. The mathematical foundation is presented thoroughly in a clear way. The topic is relevant, and the proposed methodology provides significant improvements compared to existing interpretation techniques of hydraulic tomography. Hence I only have a few minor recommendations, which I believe can further improve the manuscript.

[Figure]

General comments

The manuscript provides a very detailed methodology description, which (beyond introducing the proposed method) can be a good reference for asymptotic inversion methods used for hydraulic tomography. I found it very helpful, that the authors provided details on the existing state of the art eikonal inversion, which was later used as reference in the results sections. However, due to the shear amount of information in this section, the reader can find itself easily lost. This is why I recommend including a short overview of the proposed methodology for the end of the section (either in text or in a figure).

Beside the presented comparison with the eikonal solver, how does the proposed methodology compare to the calibration of a non-asymptotic model? Does it have any advantage, or due to the need of the h(x,t) simulation they are similar?

Specific comments

P1L7 I did not find the high permeability feature mentioned in the text, only in the abstract.

P7L30 "we need to conduct a reservoir simulation" – this is a repetition from above, consider rephrasing

P9L9 (Hu et al., 2011) recommended limiting the angles between source-receiver points before the inversion to better reconstruct layered structures with the eikonal inversion. Do you think that implying such limitations would make any difference when using the extended-trajectory-based inversion?

Hu, R., Brauchler, R., Herold, M. and Bayer, P.: Hydraulic tomography analog outcrop study: Combining travel time and steady shape inversion, J. Hydrol., 409(1–2), 350–362, doi:10.1016/j.jhydrol.2011.08.031, 2011.

P10L13 What is the reason behind choosing 10 iterations with one method and 15 with the other. What is the experience at how many iterations can the inversion be

considered complete? Are any error criteria used to determine when to stop?

P10L23 The misfit reduction associated... this sentence is too complicated while it only refers to the introduced methodology, please rephrase

P13L25 The discussion section is very brief and is mainly about the hydraulic tomography and not the presented methodology. It feels a bit odd to me, maybe consider integrating it to another section.

P14L15 What level of improvement would you expect from the proposed methodology in fractured media, where the smoothing behavior of eikonal inversion is more significant?

P14L21 mechanics

Fig. 1 – Are both open circles and filled squares used as hydraulic sources and only the circles as receivers?

Fig. 7 - By the contour lines do you mean the lines close to the top of the figure? – this is misleading in the caption. What is the role of the diagonal line leading southwest from well P3?

Fig. 11 – This reconstruction is also in very good alignment with the following papers from the same site: (Jiménez et al., 2016; Kong et al., 2018; Somogyvári and Bayer, 2017).

Jiménez, S., Mariethoz, G., Brauchler, R. and Bayer, P.: Smart pilot points using reversible-jump Markov-chain Monte Carlo, Water Resour. Res., 52(5), 3966–3983, doi:10.1002/2015WR017922, 2016.

Kong, X.-Z., Deuber, C. A., Kittilä, A., Somogyvári, M., Mikutis, G., Bayer, P., Stark, W. J. and Saar, M. O.: Tomographic Reservoir Imaging with DNA-Labeled Silica Nanotracers: The First Field Validation, Environ. Sci. Technol., acs.est.8b04367, doi:10.1021/acs.est.8b04367, 2018.

Somogyvári, M. and Bayer, P.: Field validation of thermal tracer tomography for reconstruction of aquifer heterogeneity, Water Resour. Res., 53(6), 5070–5084, doi:10.1002/2017WR020543, 2017.

Fig. 14 – Could you comment on the systematic offset of the 3 latest arrivals?

---

## Author Comment (AC2) · 7 Aug 2019

**General comments**

The reviewers comments were generally positive, in the revision we would follow their recommendations and (1) Put in a short overview of the methodology at the end of the section. (2) It is possible to construct the trajectory-based solution without a numerical reservoir simulator. That approach may be much more efficient. Even without that, the semi-analytic sensitivities have some computational advantage over numerical ones.

**Specific comments**

[Figure]

(1) P1L7. We will include that in the text.

(2) P7L30. We will remove the repetition. (3) P9L9. Such limitations may be due to experimental difficulties and might be necessary for both methods

(4) P10L13. Simply the rate of convergence. Once the method has converged no further iterations are required.

(5) P10L23. Will rephrase.

(6) P13L25. Will try and integrate it into another section.

(7) P14L15. It is hard to say what the improvement will be. A test must be conducted and a comparison made. Fractured media require the correct conceptual model, might not behave like a simple porous medium.

(8) P14L21. Will change that.

(9) Fig. 1. Locations on both sides of the model are sources and receivers.

(10) Figure 7. This is an error. The figure was changed to one without contour lines.

(11) Figure 11. Thank you for the addition references. They will be included in the revision

In summary, this review is very helpful and we will make all of the suggested changes.

---

## Author Response (AR1)

Don Vasco
Energy Geosciences Division/Building 74 - Room 204
Lawrence Berkeley Laboratory
1 Cyclotron Road
Berkeley, CA 94720
e-mail dwvasco@lbl.gov
Work (510) 486-5206
Fax (510) 486-5686

September 7, 2019

Editor: Hydrology and Earth System Sciences

Dear Editor:

I wish to submit the following revised manuscript, 'An extended trajectory mechanics approach for calculating the path of a pressure transient: Traveltime tomography' for possible publication in *Hydrology and Earth System Sciences*. In response to the editor's and reviewers comments the following changes were made:

Editor:

(1) We changed the title 'Hydraulic tomographic imaging' changed to 'Traveltime tomography'.

(2) The dimensions are provided with each new variable.

(3) Put in the word 'subsurface' on Line 10 of page 1.

(4) On page 5, around line 25 we define the scalar K in relation to the full tensor bold K.

(5) Make the various small suggested changes.

Referee 1:

(1) If someone is interested in a semi-analytic approach for imaging flow properties between boreholes then the method will be of interest to them. Such blanket statements are typical of this reviewer who was not constructive.

(2) Complete approach to inverse modeling?

a. We now compute resolution and model parameter errors, as plotted in Figures 7 and 17. We have extensive experience in model assessment in hydrology (Vasco et al. 1997), and introduced the concept of model resolution into the hydrological

community.

b. The imaging algorithm does recover the two high permeability zones. Furthermore, the application to the Widen site does produce an image that is similar to joint inversion plotted in Figure 15. As noted by Referee 2, the solution also agrees with other studies of the area.

c. This approach can be part of an integrated imaging and inversion method. The reviewer is simply wrong when they say it cannot be done. However, that was not the purpose of this paper which was to illustrate the technique and compare it to a conventional algorithm in a relatively simple situation. More complicated studies can be the topic of future papers.

d. The geometry was based upon a readily available data set from a region that has been analyzed by several investigators. The approach is general and can be applied in other settings. For example, as noted in the paper and illustrated in Vasco et al. (2000), a form of pressure travel time tomography can be applied to constant rate pumping data by using the time derivative of the transient pressure changes.

e. This is an iterative solution to the non-linear problem. We are not linearizing the problem except to take a step in our solution algorithm.

f. The Referee does not specify or define what is an acceptable inverse modeling strategy. This comment is not constructive. Regardless, we do conduct a model assessment on uncertainty quantification, if that is what the Referee is referring to.

(3) a. The core data were not extensive and, as noted in the paper, supported the general variation show in Figures 14 and 15, with higher permeabilities at shallow depths and decreasing with depth.

b. The field case is generally layered and this motivated the test case. Crosswell experiments are most accurate in quasi-two-dimensional situations. In highly three-dimensional settings there can be significant out of plane flow and transport, invalidating the approach and making a unique interpretation impossible. One is ill-advised to rely on a handful of crosswell experiments in a medium with significant out of plane heterogeneity.

c. See the discussion above.

d. 'General porous medium' means any medium that can be modeled using a reservoir simulator. For a fully 3D medium we would not use a crosswell geometry. Rather, we might try a fully 3D experiment with numerous multi-level samplers or even using multiple geophysical data sets, including geodetic data. The technique described in this paper can be used in such situations, as it is valid in a general porous medium.

e. We dispute the Referee's negative conclusion.

Referee 2:

General-

(1) We now include a brief overview at the beginning of the Methodology section, along with a schematic figure outlining the approach.

(2) In the Conclusions on page 15, Line 15, we note that the new approach only requires a forward simulation for each experiment rather then a solution of both the adjoint equation and the forward problem as is required by adjoint methods for sensitivity calculations.

Specific comments-

(1) On page 13, line 33, we note the high permeability layer.

(2) P7L30, Removed the second mention of conducting a reservoir simulation. Now page 8, line 5.

(3) P9L9. Limiting the angles may be beneficial if there are experimental issues with source receiver pairs at high angles, as in some seismic tomography work.

(4) P10L13. We now use 10 iterations for both algorithms. The number of iterations depends upon the convergence rate of each approach. This will generally depend upon how close the starting model is to the final model and the non-linearity of the inverse problem, as well as the errors on the data. Due to the modeling errors the eikonal iterations never reduce the actual reservoir simulator errors to small values, even though the eikonal traveltime errors get quite small. We stop iterating when the errors no longer decrease with successive iterations.

(5) P10L23. Edited this sentence, eliminating the unnecessary complexity. Now on page 11, Line 15.

(6) P13L25. Merged the Discussion section with the Conclusions. Now Page 14, Line 15.

(7) P14L15. If the modeling is done correctly the approach should work well in fractured media. However, I have some doubts that a continuum model will handle the fractures with sufficient accuracy.

(8) P14L21. Corrected 'mechanicss' to 'mechanics'.

(9) Fig. 1. Both are used as sources and receivers.

(10) Fig. 7. Removed the reference to contour lines. They were in a previous version of this figure.

(11) Fig. 11. Added these references to those mentioned in the discussion of this figure (now Figure 12).

(12) Yes. We have included some discussion on page 14, line 15. Because there is not much redundency in the experiment, deleting the source at position 5 may have impacted the solution, leading the large negative anomally at the right side of the model.

The reviewers have helps us to improve the paper by suggesting a figure that outlines the approach (Figure 1) and by pointing out the need for a model assessment. Though the tone of Reviewer 1 was excessively negative and most of the comments were not constructive, they did motivate us to calculate the resolution and uncertainty for both the synthetic and field cases. If there are any problems with the submission, please contact me via the above e-mail or telephone numbers. Thank you for your consideration.

Sincerely,

Don Vasco

Editor: Hydrology and Earth System Sciences

---

## Referee Report (RR1)

All my questions have been answered and all my requested revisions have been implemented into the revised manuscript. I only have a few technical comments.

P3L3 "We shall only discuss"

---

## Author Response (AR2)

Don Vasco
Energy Geosciences Division/Building 74 - Room 204
Lawrence Berkeley Laboratory
1 Cyclotron Road
Berkeley, CA 94720
e-mail dwvasco@lbl.gov
Work (510) 486-5206
Fax (510) 486-5686

October 1, 2019

Editor: Hydrology and Earth System Sciences

Dear Editor:

I wish to submit the final files for the accepted paper, 'An extended trajectory-mechanics approach for calculating the path of a pressure transient: Traveltime tomography'. Thank you, and all of the reviewers, for all of your effort.

Sincerely,

Don Vasco